# Formation of a low-mass galaxy from star clusters in a 600-million-year-old Universe

Lamiya Mowla[1,2,15 ✉], Kartheik Iyer[3,15 ✉], Yoshihisa Asada[4,5], Guillaume Desprez[4], Vivian Yun Yan Tan[6], Nicholas Martis[7], Ghassan Sarrouh[6], Victoria Strait[8,9], Roberto Abraham[10], Maruša Bradač[7], Gabriel Brammer[8,9], Adam Muzzin[6], Camilla Pacifici[11], Swara Ravindranath[12], Marcin Sawicki[4], Chris Willott[13], Vince Estrada-Carpenter[4], Nusrath Jahan[14], Gaël Noirot[4], Jasleen Matharu[8,9], Gregor Rihtaršič[7] & Johannes Zabl[4]

The most distant galaxies detected were seen when the Universe was a scant 5% of its current age. At these times, progenitors of galaxies such as the Milky Way were about 10,000 times less massive. Using the James Webb Space Telescope (JWST) combined with magnification from gravitational lensing, these low-mass galaxies can not only be detected but also be studied in detail. Here we present JWST observations of a strongly lensed galaxy at $z_{spec} = 8.296 \pm 0.001$, showing massive star clusters (the Firefly Sparkle) cocooned in a diffuse arc in the Canadian Unbiased Cluster Survey (CANUCS)[1]. The Firefly Sparkle exhibits traits of a young, gas-rich galaxy in its early formation stage. The mass of the galaxy is concentrated in 10 star clusters (49–57% of total mass), with individual masses ranging from $10^5 M_\odot$ to $10^6 M_\odot$. These unresolved clusters have high surface densities (>$10^3 M_\odot$ pc$^{-2}$), exceeding those of Milky Way globular clusters and young star clusters in nearby galaxies. The central cluster shows a nebular-dominated spectrum, low metallicity, high gas density and high electron temperature, hinting at a top-heavy initial mass function. These observations provide our first spectrophotometric view of a typical galaxy in its early stages, in a 600-million-year-old Universe.

The Firefly Sparkle is a gravitationally lensed arc identified with the Hubble Space Telescope (HST) in the CLASH survey of the galaxy cluster MACS J1423.8 + 2404 (hereafter MACS 1423) and reported as a $z > 7$ candidate[2]. Follow-up spectroscopy using MOSFIRE on the Keck telescope suggested a redshift of $z = 7.6$ based on a possible Lyman-$\alpha$ (Ly$\alpha$) detection[3]. The Canadian Unbiased Cluster Survey (CANUCS)[1] revisited the field with JWST[4], using NIRISS[5], NIRCam[6] and NIRSpec[7]. Imaging in 11 bands (0.8–5 μm) showed a long magnified arc with distinct star clusters embedded in a low surface brightness component extending up to 4′. NIRSpec Prism spectroscopy, covering the central brightest region, confirmed the high redshift ($z_{spec} = 8.296 \pm 0.001$) through multiple emission lines, with no Ly$\alpha$ emission detected. The Firefly Sparkle has two neighbours: Firefly-Best Friend (FF-BF) at $z_{spec} = 8.2996 \pm 0.0008$ and Firefly-New Best Friend (FF-NBF) at $z_{spec} = 8.2967 \pm 0.0016$. All three galaxies are shown in Fig. 1; this article focuses on the Firefly Sparkle and its star clusters.

The Firefly Sparkle resides in a highly magnified region lensed by the MACS 1423 cluster, enabling us to resolve the galaxy down to its individual star clusters. We created a magnification model using Lenstool[8,9], constrained by three multiple image systems[3] with spectroscopic redshifts from the CANUCS dataset. The model shows magnification factors between 16 and 26. A NIRSpec Prism slitlet, placed on the highest magnification region at the centre of the arc, shows strong [OIII] emission, dominating the F444W flux and making the object appear red in the composite image (Fig. 1). The projected half-light size of the arc in the source plane is only $0.3 \pm 0.1$ kpc, with most bright clusters near the centre. Eight of the ten unresolved clusters (FF-3–FF-10) are near the centre, whereas two others (FF-1 and FF-2) lie along an elongated arm, with a distance of 1.4 kpc between FF-1 and the central cluster FF-5 (all distances quoted in the paper are projected distances in the source plane). Neighbour FF-BF is even more strongly magnified ($\mu = 28.0_{-4.7}^{14.4}$), located within 2 kpc of FF-1 and also exhibits strong [OIII] emission, whereas FF-NBF is at a distance of 13 kpc with very faint [OIII] emission.

We use NIRCam and NIRISS imaging to study the resolved Firefly Sparkle and explore the stellar mass distribution in the clusters versus the diffuse arc. Photometry, derived by joint modelling of the 10 clusters and the diffuse arc using GALFIT[10], shows that nine clusters are unresolved, even in the highest resolution F115W images. Only the central cluster (FF-4) exhibits an elongated component and is fit with

[1]Whitin Observatory, Department of Physics and Astronomy, Wellesley College, Wellesley, MA, USA. [2]Center for Astronomy, Space Science, and Astrophysics, Independent University Bangladesh, Dhaka, Bangladesh. [3]Columbia Astrophysics Laboratory, Columbia University, New York, NY, USA. [4]Department of Astronomy and Physics, Saint Mary's University, Halifax, Nova Scotia, Canada. [5]Department of Astronomy, Kyoto University, Kyoto, Japan. [6]Department of Physics and Astronomy, York University, Toronto, Ontario, Canada. [7]University of Ljubljana, Department of Mathematics and Physics, Ljubljana, Slovenia. [8]Cosmic Dawn Center (DAWN), Copenhagen, Denmark. [9]Niels Bohr Institute, University of Copenhagen, Copenhagen, Denmark. [10]David A. Dunlap Department of Astronomy and Astrophysics, University of Toronto, Toronto, Ontario, Canada. [11]Space Telescope Science Institute, Baltimore, MD, USA. [12]Astrophysics Science Division, NASA Goddard Space Flight Center, Greenbelt, MD, USA. [13]NRC Herzberg, Victoria, British Columbia, Canada. [14]Shahjalal University of Science and Technology, Sylhet, Bangladesh. [15]These authors contributed equally: Lamiya Mowla, Kartheik Iyer. ✉e-mail: lmowla@wellesley.edu; kgi2103@columbia.edu

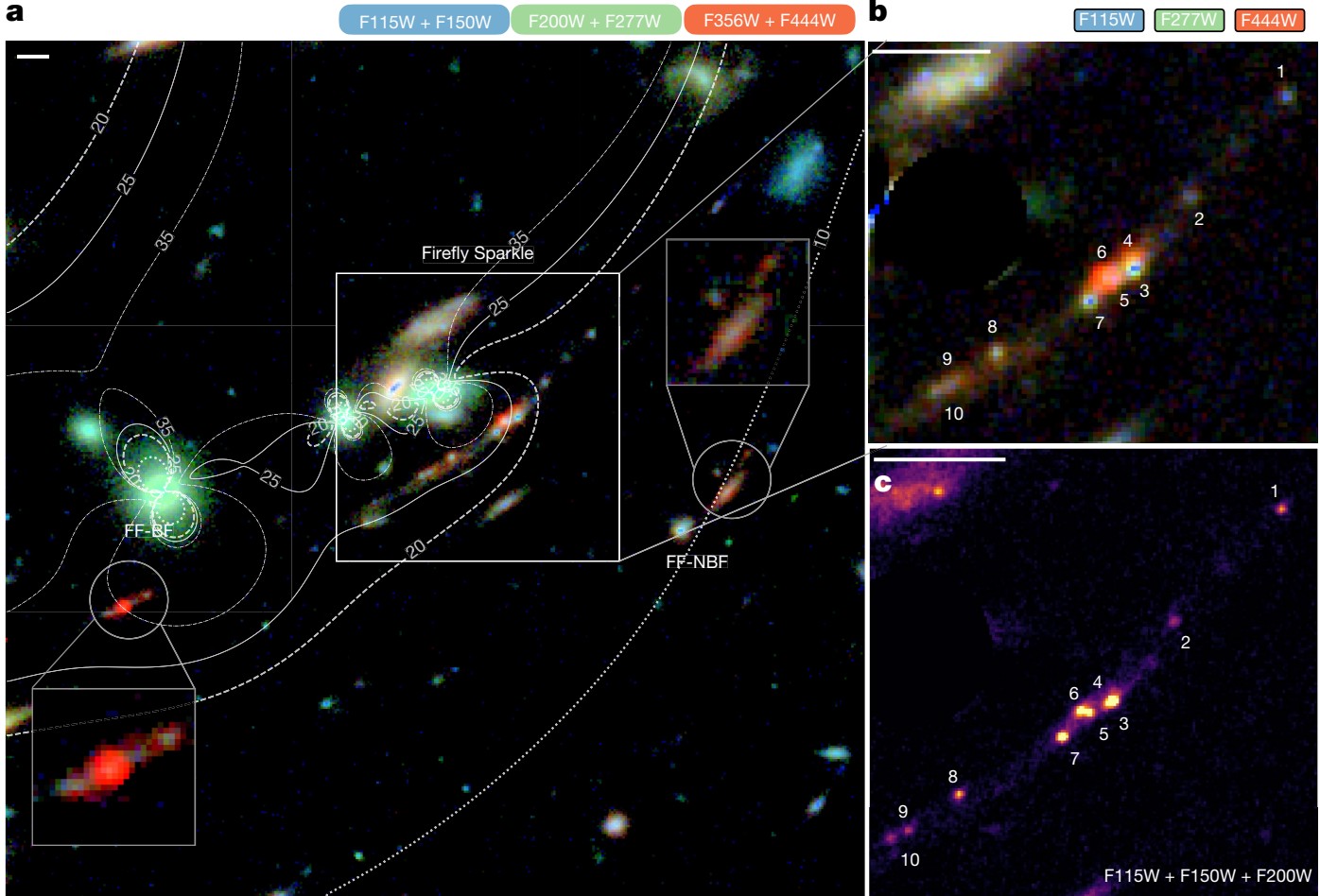

**a** F115W + F150W  F200W + F277W  F356W + F444W

**b** F115W  F277W  F444W

**c** F115W + F150W + F200W

**Fig. 1 | The Firefly Sparkle is a redshift $z_{spec}$ = 8.296 ± 0.001 gravitationally magnified arc lensed by the MACS J1423.8 + 2404 cluster. a**, Full field with the three objects of interest—Firefly Sparkle (centre), FF-BF (bottom left) and FF-NBF (bottom right)—shown in boxes and circles. The contours show the lines of lensing magnifications ($\mu$ = 15, 20, 30 and 40). **b**, RGB (F444W, F277W and F115W) image of the Firefly Sparkle showing different colours of the star clusters. **c**, Combined short wavelength (F115W + F150W + F200W) image of the Firefly Sparkle, in which the distinct clusters can be seen. Scale bars, 1″.

a Gaussian ellipse ($r_{eff}$ = 0.01). The diffuse arc is also fit with a Gaussian ellipse. All 11 components are simultaneously fit in each filter to derive the total flux (see section 'Photometry of Firefly Sparkle'). We derived an upper limit on the half-light radii ($R_{eff}$ < 0.02), which is 0.5 times the FWHM of the PSF in the F115W image, for all 10 clusters, including FF-4, whose deconvolved size is smaller than the PSF of the image. As the tangential magnifications of the clusters range from $\mu_{tan}$ = 12 to $\mu_{tan}$ = 21, this results in an upper limit on the half-light sizes of less than 4–7 pc.

We derive properties of the ionization sources in Firefly Sparkle using NIRSpec Prism spectra from two adjacent shutters (slit 1 and slit 2) covering the central brightest region of the Firefly Sparkle. Spectra and properties from slit 1 are shown in Fig. 2, whereas spectra from all slits (including BF and NBF) are shown in Extended Data Fig. 2. The spectrum of slit 1 (including light from FF-6 and contributions from FF-5 and the diffuse arc) shows a Balmer jump at $\lambda_{obs} \approx$ 3.5 μm (ref. 11) and a smooth turnover at $\lambda_{obs} \lesssim$ 1.4 μm, possibly because of two-photon continuum[12]. The absence of this feature in the spectrum at slit 2 (see section 'Photometry of Firefly Sparkle') makes damping wing of neutral hydrogen absorption unlikely. We infer significant nebular continuum contribution to the overall SED for FF-6 and possibly other clusters with similar high-EW line emission. The ionizing source effective temperature of $T_{eff} = 10^{5.1}$ K, obtained by modelling the nebular continuum with CLOUDY and from line ratios, after confirming the Balmer decrement is dust-free (see section 'Spectroscopy extraction and spectral fitting'),

suggests a hotter source than typical massive type O stars. This implies a higher upper mass limit for the IMF or a top-heavy IMF[12].

We derive the physical properties of the arc and the star clusters by performing SED fitting using various models, including simple stellar population (SSP) models and non-parametric star formation histories using the Dense Basis method (see Methods for description). SSP models are typically used in studies of star clusters in the local universe, as both observational and numerical works find that they can be approximated to single bursts, whereas other models allow for extended star formation histories. Where spectrophotometry is available, we include NIRSpec Prism spectra along with NIRISS and NIRCam photometry in the two slits (Extended Data Fig. 4), followed by photometric modelling of the properties of the individual clusters. Extended Data Table 2 shows the properties of the 10 clusters from the four different fits.

The demagnified stellar masses of the 10 clusters are about $10^5$–$10^6 M_\odot$ when fit with SSPs, similar to those of globular clusters. The surface density of the star clusters ranges between $10^3$ and $10^4 M_\odot$ pc$^{-2}$, similar to Milky Way globular clusters[13] (see Fig. 3b adapted from ref. 14). Dense Basis fits, accounting for extended star formation histories, indicate higher masses and specific star formation rates (sSFR) of about $10^{-7}$ yr$^{-1}$, showing a sharp rise in the past 10–100 Myr. The precise nature of the clusters depends on the interpretation of their star formation histories. If seen as star clusters, their masses lead to crossing times of 1–4 Myr. Combined with their age estimates, this puts them at $t_{age}/t_{cross} \sim$ 1–2,

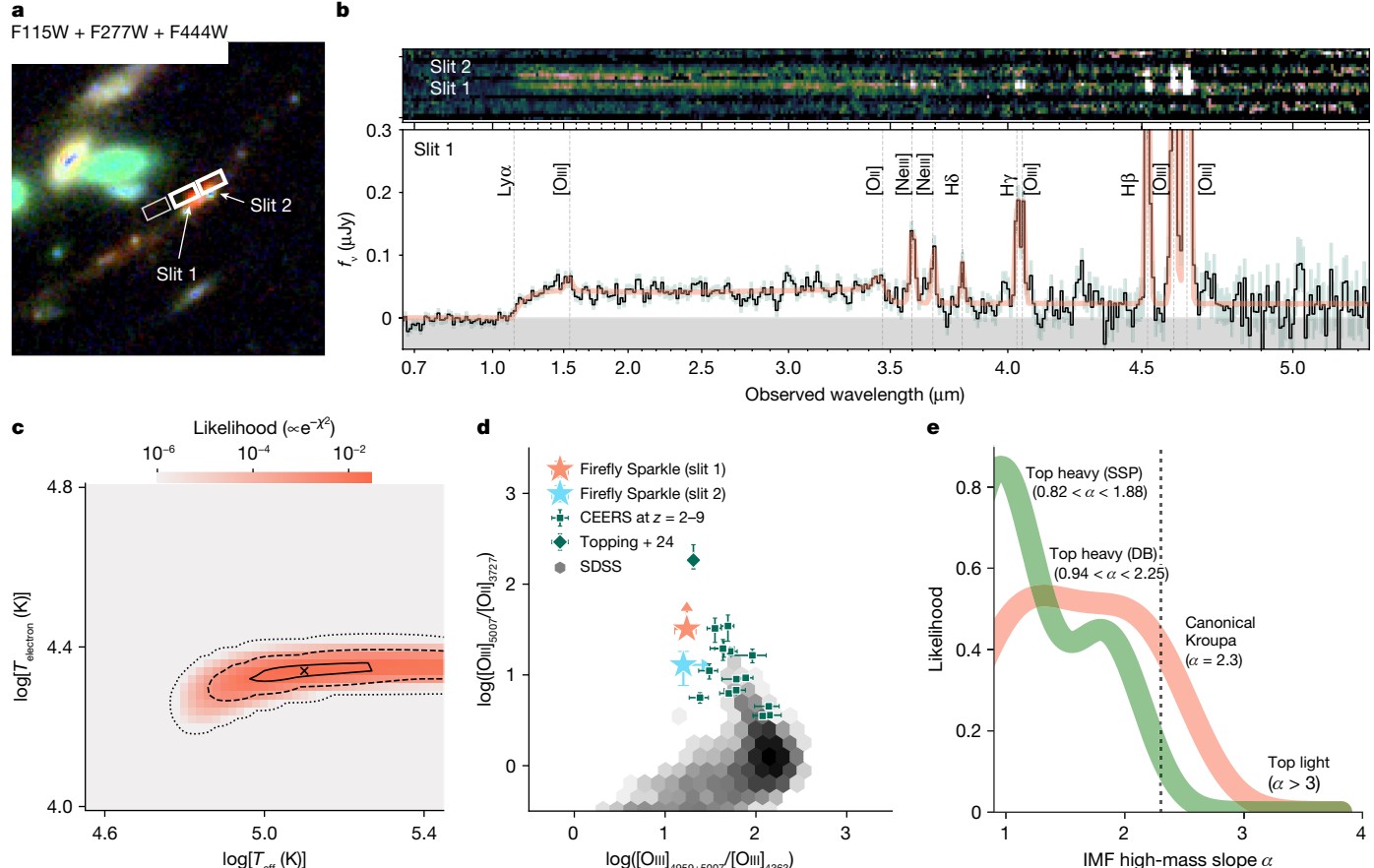

**Fig. 2 | Physical properties of the Firefly Sparkle.** The physical properties measured from the NIRSpec and Prism spectra, which include light from FF-6 as well as fractional contributions from FF-3, FF-4 and FF-5 (Fig. 1). **a**, Positions of the slitlets on the arc. **b**, The 2D NIRSpec spectrum for slit 1 (middle line, containing light from FF-6 with minor contributions from FF-5 and the diffuse arc) and slit 2 (top line, containing light from FF-4 with minor contributions from FF-3 and the diffuse arc) and the 1D spectrum of slit 1 only (see Extended Data Fig. 2 for slit 2 spectrum). **c**, Likelihood of the effective black-body temperature and electron temperature from CLOUDY modelling of the nebular continuum (1σ, 3σ and 5σ shown by contours). The slit 1 region exhibits electron temperature of $T_{electron}$ ~ 20,000 K, and ionizing source effective temperature of $T_{effective}$ ~ $10^5$ K. **d**, Emission line diagnostics estimated from the fitted line ratios for RO3 and O32 hints at a metal-poor stellar population. Error bars show 1σ uncertainties on the line ratios as derived in section 'Spectral fitting in Firefly Sparkle slit 1'. **e**, Marginal likelihood for the high-mass IMF slope from joint spectrophotometric fitting with DENSE BASIS and SSP fits suggest a top-heavy IMF (α < 2), full posteriors shown in Extended Data Fig. 4.

which indicates that they are marginally bound. By contrast, if they are nuclear star clusters or the remnants of dwarf galaxies that have previously merged with the system, their ages are consistent with having survived several crossing times, and they are likely to remain bound until ejected from the system or integrated into the nucleus. The smooth component of the arc has more demagnified mass than any individual cluster, at $\log(M_\star/M_\odot) = 6.7^{+0.9}_{-0.8}$ and an sSFR similar to the star clusters (Fig. 4). The total demagnified mass of Firefly Sparkle is $\log(M_\star/M_\odot) = 7.0^{+1.0}_{-0.3}$, one of the lowest stellar mass objects observed at this epoch, similar in stellar mass to the progenitor of a Milky Way mass galaxy at z ~ 8 (Fig. 3).

The slit 1 region shows extremely low metallicity ($\log(Z/Z_{gas}) = -0.56^{+0.13}_{-0.27}$, $12 + \log(O/H) = 7.05^{+0.22}_{-0.37}$), among the lowest observed at z > 6 (refs. 9,15,16). Our analysis of slit 1 using varying power-law slopes of the Kroupa IMF in FSPS indicates an excess of high-mass stars. Both SSP and Dense Basis fits show a preference for top-heavy IMFs in the MILES + MIST fits (Fig. 2). The high-mass star excess results in a dominant nebular continuum and high equivalent width emission lines[12]. The fits rule out top-light IMFs (α = 2.3) and prefer top-heavy slopes ($\alpha_{slit1} = 1.7^{+0.9}_{-0.7}$), consistent with a high ionizing source effective temperature of more than 40,000 K.

The current analysis has several limitations. The spectrophotometric models are influenced by star formation history, stellar population and photoionization data, affecting estimates of stellar masses, IMF and star formation histories (SFH). Future JWST observations will provide better constraints. In the meantime, we have mitigated the impact on our interpretation by using four independent SED models and focusing on aspects that are common to the models, and by providing independent measurements where possible (for example, electron temperature). Improvements to population synthesis models and refinements in the lens magnification model would help, although the estimated ages, sSFR and surface densities would be mostly unaffected.

Irrespective of these limitations, the Firefly Sparkle provides insights into the early galaxy formation. With massive star clusters exhibiting high surface density, low metallicity, high electron temperature and hints of a top-heavy IMF, the Firefly Sparkle exhibits the hallmarks of star formation in extreme environments, consistent with scenarios such as pressure-regulated feedback dominated star formation[17–19], although further observations of the gas mass are needed to ascertain this. The stellar mass of the galaxy is consistent with progenitors of Milky-Way-like galaxies, derived using the abundance matching method[20,21] and TNG50 simulation[22]. The Firefly Sparkle suggests that early galaxy assembly can occur by dense star clusters as well[23–25].

The Firefly Sparkle is the farthest spectroscopically confirmed galaxy with well-resolved star clusters, made visible by gravitational lensing

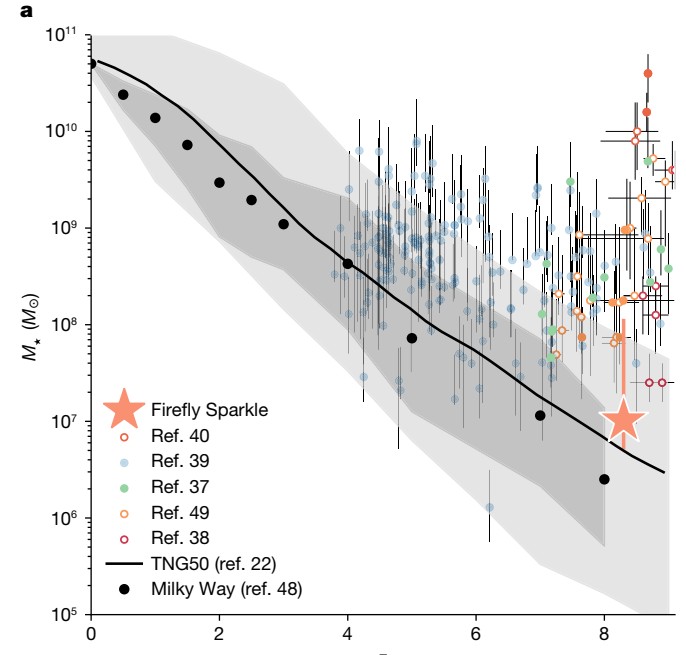

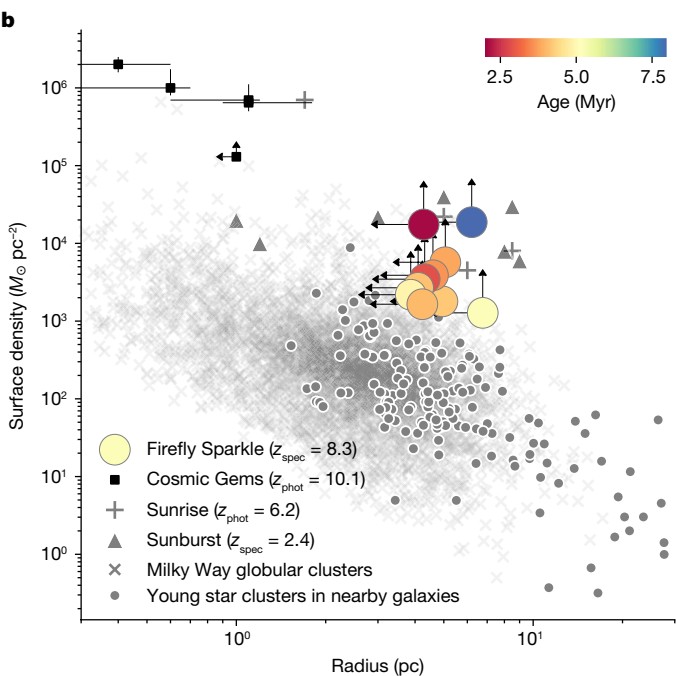

**Fig. 3 | The Firefly Sparkle in context. a**, Progenitors of Milky Way galaxy analogues in the TNG50 simulation[22] and from observations through abundance matching[20] applied to observed stellar mass functions[19,35–37]. The orange star and error bars show the median and $1\sigma$ uncertainties on the stellar mass derived from SED fitting. Given its stellar mass, the Firefly Sparkle is within $1\sigma$ of the median mass of a progenitor at $z \sim 8.3$. For comparison, stellar masses and redshifts of galaxies from previous JWST observations are shown (filled circles for spectroscopy and unfilled circles for photometry[23,37–40]). The Firefly Sparkle is one of the lowest stellar mass systems observed and the only with spectroscopically confirmed star clusters at $z > 8$. **b**, The surface density of the star clusters in the Firefly Sparkle and their sizes are shown, colour-coded by their ages from the SSP-MIST + MILES model. Nine of the ten star clusters are unresolved; hence an upper limit on the demagnified sizes (HWHM/$\mu_{tan}$) and a lower limit on the surface density are shown. Other high-redshift star clusters in magnified galaxies (Sunburst[28], Sunrise[29,41] and Cosmic Gems[14]) are shown for comparison. Milky Way globular clusters[42] and young star clusters in star-forming spiral galaxies in the Local Volume (distance less than 16 Mpc)[43] are also shown. The Firefly Sparkle star clusters are denser than local star clusters but are comparable to the higher redshift analogues.

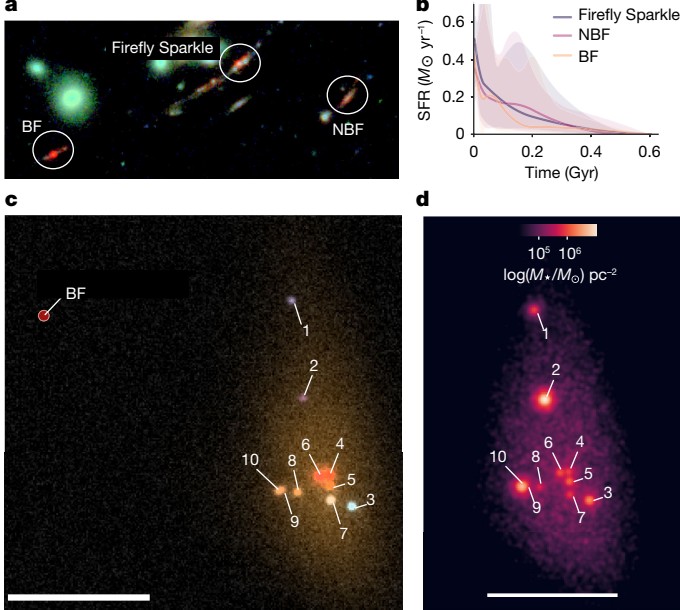

**Fig. 4 | The smooth component of the arc. a**, Image of the Firefly Sparkle, FF-BF and FF-NBF. **b**, Non-parametric integrated star-formation histories as a function of lookback time of these three galaxies reconstructed from SED fitting. The Firefly Sparkle and FF-BF both show a recent burst of star formation in the past approximately 50 Myr indicative of recent interactions. **c**, The source plane RGB (F444W, F277W and F115W) image of the Firefly Sparkle. **d**, The source plane stellar mass surface density map of the Firefly Sparkle reconstructed from the individual SED fits of the components and its source plane model; 49–57% of the total stellar mass is concentrated in the star clusters. Scale bars, 1 kpc.

and JWST sensitivity. JWST observations, combined with those of other distant galaxies[12,14,26–29], open a new area of study into the role of massive star clusters in early galaxy formation. These sites of dense and rapid star formation in distant galaxies have an uncertain future. They may survive as present-day globular clusters[30] or be stripped by tidal forces in the nascent disk of the galaxy to become nuclear star clusters[31–33]. Some clumps may even survive tidal stripping and loss, as simulations suggest that they can reaccrete gas in the turbulent environment[34]. Future observations by JWST and ALMA will help distinguish these possibilities.

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

# Methods

## Image preparation

The cluster field MACS J1423.8 + 2404 was observed with JWST/NIRCam imaging using filters F090W, F115W, F150W, F200W, F277W, F356W, F410M and F444W with exposure times of 6.4 ks each, reaching a signal-to-noise ratio between 5 and 10 for an $m_{AB}$ = 29 point source. It was also observed with JWST/NIRISS imaging using filters F115W, F150W and F200W.

To reduce the imaging data, we use the photometric pipeline that is presented in more detail in ref. 44. Briefly, the raw data has been reduced using the public grism redshift and line analysis software Grizli[43], which masks imaging artefacts, provides astrometric calibrations based on the Gaia Data Release 3 catalogue[13] and shifts images using Astrodrizzle. The photometric zero-points are applied as described in ref. 34. RGB image created using six filters of NIRCam observation of the Firefly Sparkle is shown in Fig. 1. We used images from which bright cluster galaxies and intracluster light have been removed, as described in ref. 25. The methodology for modelling and removing diffuse light from cluster galaxies and intracluster light (ICL) is presented in ref. 25. The NIRCam depths (0.3′ diameter aperture) for F090W, F115W, F150W, F200W, F277W, F356W, F410M and F444W are 7.2, 6.6, 5.2, 4.4, 3.0, 2.9, 5.5 and 4.3 nJy, respectively, and the NIRISS depths for F115WN, F150WN and F200WN are 3.6, 4.3 and 4.0 nJy, respectively[41].

## Photometry of Firefly Sparkle

We perform photometry in 10 JWST bands (NIRISS: F115WN, F150WN and F200WN; NIRCam: F115W, F150W, F200W, F277W, F356W, F410M and F444W) in which the Firefly Sparkle is detected from their morphological fit with GALFIT. In other JWST and HST filters, the Firefly Sparkle is not or barely detected; hence, we place upper limits for the entire source. As the object is resolved into at least 10 distinct clusters and a diffuse galaxy component, we perform a morphological fit using Galfit[10] to extract the photometric information.

Point spread functions are extracted empirically by median stacking bright, isolated, non-saturated stars following the methodology described in ref. 28. Convolution kernels for homogenizing all data to the F444W resolution are created with photutils.psf.matching using a SplitCosineBellWindow() windowing function to remove high-frequency noise, which results from floating-point imprecision when taking the ratio of Fourier transforms. We optimize the shape of each window function to minimize the median residual between convolved stars from each source filter that is convolved and stars from the target F444W filter.

For the morphological fit, we create 10″ × 10″ postage stamps in all 10 filters from the BCG-subtracted images. We determine the priors for the centres of the 10 clusters by visual inspection. Although nine out of the ten appear as point sources, FF-4 has an elongated shape and appears unresolved. We first determine the central coordinates of the 10 clusters and the arc by fitting (1) an elliptical Gaussian for FF-4; (2) nine point sources for the other nine clusters; and (3) another elliptical Gaussian with the bending mode turned on for the diffuse arc to the F115W image, which has the highest resolution (smallest PSF). The free parameters are the centres and total fluxes of all the components, the radius and axis ratio of FF-4, and the radius, axis ratio and bending mode (B2) of the arc. The initial guesses for the coordinates were determined by visual inspection of the F115W image. Once we obtain the fitted central coordinates of all the components from F115W, we again fit all 11 components in F444W, which has the highest signal-to-noise ratio for the arc and FF-4, to determine the radius, axis ratio, position angles of the ellipses, and the bending mode B2 of the arc.

We use the best-fit centre coordinates from F115W as the central coordinates in all the filters. However, instead of fixing the central coordinates, we allow GALFIT to fit for them in every filter within a very narrow range of ±0.5 pixels (0.02″) to account for the uncertainty in the PSF centre. We also fix the bending mode B2 (2.14), ellipse radius (3.9″), axis ratio (0.08) and position angle (−51.8°) of the arc from the F444W fit. We fix the morphology of FF4 also with radius = 0.59″, axis ratio = 0.1 and position angle = −53°.

We now fit all 11 components in all 10 filters to determine their fluxes. The resulting models and residuals are shown in Extended Data Fig. 1. Residuals from the fits are negligible, as shown by $\chi^2/\nu \sim 1$ in the GALFIT fits in all filters. This confirms the original visual impression that nine of the ten clusters are unresolved and an additional smooth component is present.

To derive the uncertainty in our flux estimation, we inject the full Firefly Sparkle model in 100 random locations in our 10″ × 10″ postage stamps (avoiding the edge) and refit with the exact same setting of GALFIT. We find no significant systematic offset between the fitted flux and the injected flux for any of the 11 components, in any of the filters, showing that our photometric technique is robust to background variations across all filters. The uncertainty in the photometry is calculated from the bi-weight scale of the 100 refitted fluxes. The resulting photometry and the RGB image of the model and the residual are shown in Extended Data Fig. 1. The agreement between NIRISS and NIRCam fluxes in the three overlapping filters is another confirmation of the robustness of photometry. We have used updated zero-points[34] and corrected for Milky Way extinction using the colour excess $E(B − V)$ = 0.0272 from ref. 6 and assuming the extinction law in ref. 35 using the factor between the extinction coefficient and colour excess $R_V$ = 3.1.

## Spectroscopy extraction and spectral fitting

NIRSpec spectroscopy has been acquired for MACS J1423.8 + 2404 and spectra were obtained for the Firefly Sparkle, FF-BF and FF-NBF. The spectra for the FF-BF were part of the sample in ref. 23, with $z_{spec}$ = 8.2953 ± 0.0005. The spectra were observed using the PRISM/CLEAR disperser and filter, through three Micro-Shutter Assembly (MSA) masks per cluster with a total exposure time of 2.9 ks per MSA configuration.

The NIRSpec data were processed using the STScI JWST pipeline (software v.1.8.4 and jwst_1030.pmap) and the msaexp package[31]. We used the standard JWST pipeline for the level 1 processing, in which we obtained the rate fits files from the raw data. We enabled the jump step option expand_large_events to mitigate contamination by snowball residuals and used a custom persistence correction that masked out pixels that approach saturation within the following 1,200 s for any readout groups. We then used msaexp for level 2 processing, for which we performed the standard wavelength calibration, flat-fielding, path-loss correction and photometric calibration and obtained the 2D spectrum before background subtraction. As the central and upper shutters contain different clusters (see Fig. 2a to find the shutter positions), we need custom background subtraction to avoid self-subtraction. We did this by building the background 2D spectrum by stacking and smoothing the sky spectrum in the empty pixels and obtained the background subtracted 2D spectrum of Firefly Sparkle. We confirmed that this custom background subtraction method works as well as a standard drizzle background subtraction method used in the literature[33], using a well-isolated galaxy spectrum from the CANUCS observation (Asada et al., in prep.). We finally extract the 1D spectrum separately in slit 1 and slit 2, by collapsing the 2D spectrum using an inverse-variance weighted kernel following the prescription in ref. 24. We verified that the uncertainty array of the 1D spectrum has the appropriate normalization by testing the distribution of spectral fluctuations in an empty sky region and finding the fractions of pixels at >1 and >2$\sigma$ as expected.

**Spectral fitting in Firefly Sparkle slit 1.** The resulting 1D spectrum of Firefly Sparkle in slit 1, dominated by the cluster FF-6, is shown in Fig. 2.

The spectrum exhibits a Balmer jump at $\lambda_{obs} \sim 3.5$ μm and a turnover at $\lambda_{obs} \lesssim 1.4$ μm, probably because of two-photon emission. These

features suggest that the nebular continuum should dominate over the stellar continuum in the rest frame UV to optical spectrum within slit 1 (as found for a $z = 5.9$ galaxy in ref. 12). We thus model the continuum of the spectrum with nebular continuum using the photoionization code CLOUDY v.23 (ref. 5). To determine the dust attenuation value in the continuum model fitting, we first measure the Hγ/Hβ ratio by fitting the Gaussian profiles. The ratio agrees well with the case B recombination, and no significant dust attenuation is indicated. Therefore, in the continuum spectral modelling, we use pure hydrogen gas irradiated by an ionizing source having black-body SED without dust attenuation. We vary the effective temperature of the black body ($T_{eff}$) and the electron temperature of the (ionized) hydrogen gas ($T_{e,H^+}$), and search for the best-fitting model continuum by $\chi^2$ minimization. In the continuum fitting, we mask out emission line regions and all wavelengths $\lambda_{obs} < 1.2\,\mu m$ at which the Lyman break is seen in the slit 2 spectrum, because this region may be affected by a neutral hydrogen damping wing. The best-fit model has $\log(T_{eff}/K) = 5.10$ and $\log(T_{e,H^+}/K) = 4.34$, which is fully consistent with the results in ref. 12. The result of continuum fitting does not change if we consider a slight dust attenuation ($A_V = 0.1$ mag) in the fitting. As discussed in ref. 12, the effective temperature of $\log(T_{eff}/K) = 5.10$ is much hotter than typical massive type O stars and is suggestive of this star-forming cluster having a top-heavy IMF. The IMF of this cluster is further discussed in section 'SED fitting analysis'.

Note that the UV continuum turnover feature could be because of the absorption from dense neutral hydrogen either in the intergalactic medium (IGM) or in the circumgalactic medium (CGM). However, in the case of slit 1 spectrum, we expect the effect of IGM and CGM damping absorption to be negligible or limited at $\lambda_{obs} < 1.2\,\mu m$ based on the blue continuum and sharp drop-out in the slit 2 spectrum (see section 'Spectral fitting in Firefly Sparkle slit 2' for details of slit 2 spectrum). Considering the spatial proximity of the slit 1 and slit 2 regions (Fig. 2), we can assume the absorption feature from line-of-sight neutral hydrogen to be the same in the slit 1 and slit 2 spectra. The slit 2 spectrum is rather blue and has a sharp Lyman break starting at $\lambda_{obs} = 1.2\,\mu m$, whereas the slit 1 spectrum shows the turnover starting at $\lambda_{obs} \sim 1.4\,\mu m$. Thus, the turnover feature should not be because of the neutral hydrogen absorption, but rather because of the intrinsic continuum shape of the source. Nevertheless, to avoid the possible effect of the neutral hydrogen absorption, we mask out $\lambda_{obs} < 1.2\,\mu m$ in the nebular continuum fitting above (corresponding to 1,290 Å in the rest frame).

Having the model continuum, we subtract the underlying model continuum from the observed spectrum and measure the spectroscopic redshift and emission line fluxes by fitting Gaussian profiles. The best-fitting model spectrum with nebular continuum and Gaussian profiles is shown in Fig. 2b (red solid curve). We securely detect emission lines of [O III]$\lambda\lambda$4959, 5007, Hβ, [OIII]$\lambda$4363, Hγ, Hδ and [NeIII]$\lambda\lambda$3869, 3889. We do not find significant detection of [OII]$\lambda$3727 and obtain an upper limit for the flux of this line. There is a tentative detection of the blended line of [OIII]$\lambda\lambda$1661 + 1666, although the spectral resolution of the prism is low at this wavelength making this doublet difficult to securely detect and separate from HeII$\lambda$1640. We use these emission line fluxes to estimate the physical parameters in slit 1. We first estimate the dust attenuation based on Balmer decrements. Both the Hγ/Hβ and Hδ/Hβ ratios are consistent with theoretical predictions in case B recombination[21] within the uncertainties, suggesting there is no significant dust attenuation (Extended Data Fig. 3, red squares in the left). This result is consistent with the initial measurement before the continuum fitting above and supports the validity of the dust-free assumption in the nebular continuum fitting process. Therefore, we do not correct for dust attenuation in the following measurements of emission line ratios and physical parameters in this section.

We next measure the electron temperature using temperature-sensitive emission line ratios: [OIII]$_{4959+5007}$/[OIII]$_{4363}$ and [OIII]$_{5007/}$

[OIII]$_{1661+1666}$. We assume the electron density to be $n_e = 10^3\,cm^{-3}$, which is consistent with recent JWST observations of similarly high-$z$ galaxies[7] and obtain consistent independent temperature measurements within the uncertainties ($T_{e,O^{++}} = 4.0^{+2.6}_{-0.9}$ K and $2.9^{+0.7}_{-0.4} \times 10^4$ K, respectively; Extended Data Fig. 3 (right)). Note that because the [OIII]$\lambda\lambda$1661 + 1666 detection is tentative and potentially blended with HeII$\lambda$1640, we consider [OIII]$\lambda$4363 to be more reliable.

We note that in ref. 16, the authors measured a similar ratio of [OIII]$_{4959+5007}$/[OIII]$_{4363}$ in the $z = 6$ galaxy RXCJ2248-ID to that of slit 1. In ref. 16, medium resolution spectroscopy was used to determine the electron density directly. They found that when using lines with higher ionization potential than O+, the electron density was higher ($n_e \sim 10^5\,cm^{-3}$) than is typically found from [OII]$\lambda$3727 (ref. 7). This high electron density leads to a lower electron temperature for their galaxy of $T_{e,O^{++}} = 2.5 \times 10^4$ K. Similarly, if we assume the electron density of $n_e = 10^5\,cm^{-3}$ instead for our slit 1 spectrum, the electron temperature from [OIII]$\lambda$4363 becomes $T_{e,O^{++}} = 3.2^{+1.6}_{-0.96}$, which is in between the two measurements based on [OIII]$\lambda\lambda$1661 + 1666 and [OIII]$\lambda$4363 when assuming $n_e \sim 10^3\,cm^{-3}$ above. To consider the possibility of a somewhat higher electron density in the highly ionized region, we adopt the mean value of our two electron temperature measurements ($T_{e,O^{++}} = 3.5 \times 10^4$ K) as our fiducial value and propagate the full range of the two measurement uncertainties into the following metallicity measurement.

Based on the electron temperature measurement, we obtained the oxygen abundance from [OIII]$_{4959+5007}$/Hβ and [OII]$_{3727}$/Hβ ratios, following the prescription in ref. 8. We assume the electron density to be $n_e = 10^3\,cm^{-3}$. The total oxygen abundance is calculated from O$^{++}$/H$^+$ and O$^+$/H$^+$, and the higher ionizing state oxygen is ignored[30]. As the [OII]$\lambda$3727 emission line is undetected, we can obtain only an upper limit for O$^+$/H$^+$, but the upper limit for the abundance of the singly ionized oxygen is negligibly small as compared with the doubly ionized oxygen. We thus derived the total oxygen abundance from O$^{++}$/H$^+$, yielding $12 + \log(O/H) = 7.05^{+0.22}_{-0.37}$ ($Z_{gas}/Z_\odot = 0.02^{+0.04}_{-0.01}$ assuming the solar abundance to be 8.69; ref. 38).

We also derive the ionization parameters using the ionization-sensitive emission line ratios: [OIII]$_{5007}$/[OII]$_{3727}$ and [NeIII]$_{3869}$/[OII]$_{3727}$. Following the prescription in refs. 45,46, we obtain the lower limit for the ionization parameters ($\log U$) from these two ratios. Both ratios provide a similar limit of $\log U > -2.0$.

All the emission line flux measurements and the derived physical parameters in Firefly Sparkle slit 1 are presented in Extended Data Table 1. We also compare the diagnostic emission line ratios in Firefly Sparkle with those in other galaxy population in Fig. 2d. We use the ionization-sensitive line ratio O32 ([OIII]$_{5007}$/[OII]$_{3727}$) and the temperature-sensitive line ratio RO3 ([OIII]$_{4959+5007}$/[OIII]$_{4363}$) and compare these line ratios with other [OIII]$\lambda$4363-detected galaxies at $z = 2$–9 from previous JWST observations[2] and those in the local universe from SDSS observations[14]. Extended Data Fig. 3 (middle) presents a similar comparison but uses another ionization-sensitive line ratio Ne3O2 ([NeIII]$_{3869}$/[OII]$_{3727}$) instead of O32.

**Spectral fitting in Firefly Sparkle slit 2.** In contrast to slit 1, the extracted 1D spectrum in Firefly Sparkle slit 2 does not show nebular continuum features, and the blue continuum is rather smooth with a sharp drop-out because of the Lyman break at $\lambda_{obs} \sim 1.2\,\mu m$. We thus derive the emission line fluxes from the slit 2 spectrum by fitting Gaussian profiles with the continuum being modelled by a constant offset around each emission line. We detect [OIII]$\lambda\lambda$4959,5007, Hβ, Hγ, Hδ, [NeIII]$\lambda$3869 and [OII]$\lambda$3727 emission lines in the slit 2 spectrum but do not detect [OIII]$\lambda$4363.

We then derive the physical properties in the same way as done for Firefly Sparkle slit 1 spectrum. We measure the dust attenuation from Balmer decrement, Hγ/Hβ and Hδ/Hβ, and find both line ratios agree well with the predicted ratios under case B recombination (blue squares

in Extended Data Fig. 3 (left)). This suggests that the dust attenuation is negligible in the slit 2 spectrum as well, and we do not make a dust attenuation correction.

As we do not detect the temperature-sensitive emission lines of [OIII]$\lambda$1666 or [OIII]$\lambda$4363 in the slit 2 spectrum, we cannot measure the electron temperature and the metallicity from the direct-temperature method. We thus obtain only the upper limit for the electron temperature ($T_{e,O^{++}}$) from the non-detection of [OIII]$\lambda$4363. The electron temperature in Firefly Sparkle slit 2 is shown to be $T_{e,O^{++}} < 1.8 \times 10^4$ K (1$\sigma$) or $<4.5 \times 10^4$ K (3$\sigma$). To visualize the difference in physical properties in slit 1 and slit 2, we show the diagnostic emission line ratios of Firefly Sparkle slit 2 in Fig. 2d and Extended Data Fig. 3 (middle) as well.

## SED fitting analysis

SEDs derived from our photometry were analysed using a slightly modified version of the Dense Basis method[18,47] to determine non-parametric SFHs, masses and ages for our sources in Firefly Sparkle. We adopt the Calzetti attenuation law[48] and a Kroupa IMF[32] with a flat prior for the high-mass slope $\alpha \in [1., 4.]$. We run fits using both the MILES stellar libraries[29] and MESA Isochrones and Stellar Tracks (MIST; ref. 17), as well as the Binary Population and Spectral Synthesis (BPASS; refs. 26,36) models to consider for the presence of binary populations. As the latest BPASS version in FSPS (-bin-imf135all 100) assumes a Salpeter IMF with an upper mass cutoff of $100 M_\odot$ and does not allow for a varying IMF, we only vary the top-heavy slope of the Kroupa IMF in the MILES + MIST runs with an upper mass cutoff of $120 M_\odot$. This should be considered while comparing the physical properties from the two runs, as allowing for a varying IMF based on the MILES + MIST configuration results in lower stellar masses for those runs because they are preferentially fit with top-heavy SSPs with a greater fraction of light coming from more massive stars. We fix the redshift to that found from the NIRSpec Prism spectroscopy by the [OIII] $\lambda$4959 line at $z_{spec} = 8.296 \pm 0.001$. All other parameters are left free. We run the SED fits in two configurations to account for different possibilities of the nature of the individual clusters:

1. SSP fits: to account for the possibility that the individual clumps are star clusters, which is likely given the physical scales of the clusters and the extreme emission lines in the spectra, we modify the code to fit for instantaneous bursts of star formation, described by SSPs. In this case, we assume a flat prior in the log age of the SSP from $10^5$ years to the age of the universe at $z_{spec} = 8.296 \pm 0.001$ instead of the non-parametric defaults for the SFH in Dense Basis.

2. Non-parametric SFH fits (Dense Basis): to fit the diffuse body of the galaxy and to account for the possibility that the clusters are nuclear star clusters or remnants of previous mergers, we also run fits with non-parametric SFHs with a Dirichlet prior. The main advantage of using Dense Basis with non-parametric SFHs is that they allow us to account for flexible stellar populations, which is important at these redshifts[49] because star formation is expected to be stochastic and may be underestimated if fit using traditional parametric assumptions[39,50].

We perform our fitting in two stages—we initially perform a joint spectrophotometric fit to the NIRSpec Prism spectrum along with the HST + NIRISS + NIRCam photometry in the slits in which both exist (Extended Data Fig. 4). We correct for slit loss considering two factors—the amount of light lost due to the changing PSF as a function of wavelength and an overall multiplicative correction to match the spectrum against the photometric measurements. We modify the default Dense Basis method in this stage to additionally fit for the slope at the massive end of the IMF, the gas-phase metallicity and the ionization parameter, using the relevant parameters from FSPS (imf3, gas_logz and gas_logu). Doing so allows us to substantially constrain priors on star formation rate, IMF, dust, ionization parameter and metallicity

that we then use to fit the photometry. We find that the fits are consistent with negligible dust attenuation, consistent with our estimates from measuring the Balmer decrement. We also find that our fits rule out the part of parameter space consistent with the canonical Chabrier-like or Kroupa-like IMF (with the high-mass slope ≈ 2.3) in favour of more top-heavy slopes of about $1.5^{+0.7}_{-0.6}$ for slit 1, which contains portions of clusters 3, 4, 5 and 6. We find weaker constraints from the spectrum for slit 2, which still skews towards top-heaviness but with large uncertainties of about $1.7^{+0.9}_{-0.7}$. Finally, we find estimates of both stellar and gas-phase metallicities to be sub-solar, consistent with estimates from the line ratios.

Using our photometry (Extended Data Table 4), we now determine the stellar properties of each individual component by running a second set of fits using the same set of parameters that are used to fit the spectrophotometry. Although parameters such as the metallicity and ionization parameter are only loosely constrained by these fits, we obtain parameter estimates for the stellar masses, star formation rates and ages of the individual star clusters with uncertainties that marginalize over the variations in the other parameters.

Both photometry and corresponding fits to the SED fit are shown in Extended Data Fig. 5, with variations in the stellar mass, age and reduced $\chi^2$ of the fits for each of the four scenarios (SSPs fits with MILES + MIST and BPASS, and Dense Basis fits with MILES + MIST and BPASS) shown in Extended Data Table 2. All 10 components have intrinsic (corrected for magnification) stellar masses of about $10^5$–$10^6 M_\odot$ and sSFR of $10^{-7}$ yr$^{-1}$. Although the error bars are large, the distinct colours of the clusters hint at different formation times. Although the smooth component contains a large fraction of the stellar mass, the bulk (about 57%) lies in the clusters. Extended Data Table 3 lists the physical properties of the individual components as well as the full Firefly Sparkle, BF and NBF galaxies.

We find that the SSP fits are generally less massive compared with the Dense Basis fits, because the light from the SED is modelled by a single epoch of star formation instead of an extended episode. As light from the massive stars responsible for young star formation are much brighter than older stellar populations, they can describe the observed SED with a lower mass. However, the SSP fits often cannot capture both the UV slopes and the nebular emission in the rest-optical, as seen for clusters 1, 3, 7 and 8 in Extended Data Fig. 5 and often approximate it using a Balmer break, leading to posteriors consistent with much older ages than the median values.

Although the $t_{age}$ from SSP and $t_{50}$ from Dense Basis fits (Extended Data Table 2) may seem inconsistent, it is important to note that the Dense Basis fits for most star clusters indicate a sharp burst of star formation within the past 10 million years (Extended Data Fig. 6). By design, an SSP is biased towards this recent burst, whereas a non-parametric SFH can accommodate extended episodes of star formation. However, with our current data, we cannot distinguish between extended SFH in the star clusters and the contribution of light from the diffused arc.

The masses of the clusters also scale with the top-heaviness of the high-mass end of the IMF in the MILES + MIST fits, with lower masses for more top-heavy IMF values as that scale the amount of light from massive stars. In comparison, the BPASS fits in the current setup are done at the canonical Kroupa IMF, leading to higher masses for those fits. At the same IMF slope, the masses are comparable within uncertainties for the different SPS models, and the sSFR and age/$t_{50}$ values are consistent even marginalizing over the IMF posteriors. Given the observational constraints and the $\chi^2$ from the fits in Extended Data Table 2, it is not currently possible to definitively rule out any of the current fitting approaches.

## Lens modelling

We use Lenstool[9] to build a strong lensing model of the MACS 1423 cluster, to be fully presented in Desprez et al. (manuscript in preparation).

This model is constrained with the three multiple image systems that were leveraged in ref. 3, for which we provide additional information obtained from the CANUCS data. The first two systems are those presented in ref. 27, one at $z = 2.84$ for which we account for the two clusters visible in the four images of the objects, and the second one with three images at $z = 1.779$ for which we identify another cluster in the two northernmost images for improved constraints. The last system is composed of five images[11] for which we provide a new spectroscopic redshift measurement of $z = 1.781$ that is in agreement with photometric and geometric redshifts previously measured.

The different mass components are parameterised as double Pseudo-Isothermal Elliptical (dPIE) profiles[4]. The model is composed of a large cluster scale mass halo, an independent galaxy scale centred on the brightest cluster galaxy and small galaxy scale mass components to account for the contribution of all cluster members that follow a mass–luminosity scaling relation[22]. For all galaxies, their positions, ellipticities and orientations have been fixed to these measured from the images. The final best model manages to reproduce the position of the input multiple images with a distance rms of 0.46″.

Magnifications are obtained by generating convergence and shear maps around the Firefly Sparkle with a size of 20″ and a resolution of 10 milli-arcsec per pixel. Uncertainties in the magnifications are computed from 100 randomly selected models from the optimization of Lenstool after its convergence around the minimum $\chi^2$. The numbers provided in Extended Data Table 3 are the median and $\pm 1\sigma$ limits on the distribution of the 100 values obtained at the position of each cluster. We measured the average magnification of the FF-arc by using the GALFIT model of the arc (in F200W) and selecting all pixels with flux >10% of maximum flux. We then computed the best magnification value for all selected pixels and computed the mean and standard deviation values for these to find the magnification of the arc ($\mu = 24.4 \pm 6.0$).

The source plane reconstruction is made using the best GALFIT model to compute the source plane positions and magnification for the 10 star clusters. We use Lenstool to generate a source plane image reconstruction of the diffuse light of the galaxy with a smooth PSF-deconvolved model of its light profile. We use GALFIT to add 10 point sources convolved with the appropriate PSFs to the diffused source plane model at the source plane positions of the star clusters with the demagnified fluxes. This process is repeated to generate source plane models in all filters. We also generate a mass map using the same prescription, replacing the demagnified fluxes with the demagnified masses. The resulting source plane RGB image and mass map are shown in Fig. 4c,d.

## Size and surface density of star clusters

We now investigate the spatial properties of the star clusters. Nine out of the ten star clusters are unresolved even in our highest resolution F115W NIRCam image. FF-4 has a slightly elongated shape visually but has a best-fit major axis size (0.01) smaller than the smallest PSF, making the size estimate unreliable. Hence, we use the half-width half-max of the NIRCam F115W PSF (0.02) to set an upper limit on the size of all 10 star clusters. To determine the upper limits of the sizes of unresolved sources, we use the tangential eigenvalue of magnification $1/|\lambda_t|$, which ranges between 14 and 24. This results in a size upper limit between 4 pc and 7 pc. The central star clusters have the highest magnification and the smallest upper limits, whereas the ones near the two ends of the arc have the lowest. We use the upper limit on sizes and the demagnified stellar masses to calculate the lower limit on stellar surface densities as shown in Fig. 3b.

## Abundance matching for MW and M31 progenitors

To estimate the range of stellar masses of progenitors of both MW-mass and M31-mass galaxies at higher redshift, we adopt a semi-empirical approach combining both simulations and observations. We assume an evolving co-moving number density with redshift, as determined by the abundance matching code from ref. 20, with $z = 0$ number densities of $\log(n/\mathrm{Mpc}^3) = -2.95$ and $\log(n/\mathrm{Mpc}^3) = -3.4$, respectively, for MW and M31 mass analogues. The code calculates a past median galaxy number density at $z_2$, given an initial number density at $z_1$, using peak halo mass functions. As the merger rate per unit halo per unit $\Delta z$ is roughly constant, the evolution of the cumulative number density of progenitors of any given galaxy is a power law, with the change described by $(0.16\Delta z)$ dex.

In ref. 20, peak halo mass functions are used because the resultant median number densities are less affected by the scatter in stellar mass and luminosity. However, this scatter does affect the $1\sigma$ errors in cumulative number density. The $1\sigma$ or 68 percentile range grows with increasing redshift, but this growth is also higher for more massive galaxies.

As the code from ref. 20 does not calculate stellar masses, we obtain the stellar mass ranges of the progenitor populations of MW and M31 analogues using stellar mass functions (SMFs) from various surveys[15,19,40]. We take the median cumulative number densities at each $\Delta z$ to find the stellar mass associated with that number density from the corresponding SMF. Moreover, the $1\sigma$ errors on the given number density for each redshift are then used to determine the $1\sigma$ errors on the stellar mass of the progenitors. At $z = 8.3$, the median stellar mass of MW progenitors is $\log(M_\star/M_\odot) = 6.4 \pm 0.7$ and the median stellar mass of M31 progenitors is $\log(M_\star/M_\odot) = 6.9 \pm 0.8$. The Firefly Sparkle with a stellar mass of $\log(M_\star/M_\odot) = 7.0^{+1.0}_{-0.3}$ is definitely within $1\sigma$ stellar mass range of both Milky Way and M31 progenitors. More details on the progenitor matching technique can be found in ref. 37.

## Data availability

All data supporting the findings of this study are publicly available on the CANUCS website at GitHub (https://niriss.github.io/). Further requests for data can be directed to the corresponding authors.

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

**Acknowledgements** We thank N. Murray for the discussions about the formation of star clusters and their dynamic state, and E. Nelson, J. Antwi-Danso, G. Bryan and R. Somerville for their discussions. We thank N. Gosavi for coining the name Firefly and O. Trottier for his help with Fig. 4. This research was funded by grant 18JWST-GTO1 from the Canadian Space Agency and the Natural Sciences and Engineering Research Council of Canada. L.M., K.I. and R.A. acknowledge the support of the Dunlap Institute for Astronomy and Astrophysics at the University of Toronto. K.I. was supported by NASA through the NASA Hubble Fellowship grant HST-HF2-51508 awarded by the Space Telescope Science Institute, which is operated by the Association of Universities for Research in Astronomy, for NASA, under contract NAS5-26555. Y.A. was supported by a Research Fellowship for Young Scientists from the Japan Society of the Promotion of Science (JSPS). M.B. and G.R. acknowledge support from the ERC Grant FIRSTLIGHT and from the Slovenian National Research Agency ARRS through grants N1-0238, P1-0188 and the program HST-GO-16667, provided through a grant from the STScI under NASA contract NAS5-26555. This research used the Canadian Advanced Network For Astronomy Research (CANFAR) operated in partnership by the Canadian Astronomy Data Centre and The Digital Research Alliance of Canada with support from the National Research Council of Canada, the Canadian Space Agency, CANARIE and the Canadian Foundation for Innovation.

**Author contributions** L.M. conducted the photometry and size measurements, K.I. performed the SED fitting, Y.A. conducted the spectral analysis, G.D. conducted the lens modelling and V.Y.Y.T. performed the progenitor mass analysis. L.M., K.I., Y.A., G.D. and V.Y.Y.T. also made the figures and wrote the paper. G.B., C.W. and V.S. were responsible for the image processing pipeline and generating image mosaics. N.M. conducted the BCG subtraction and G.S. performed the PSF modelling. All authors made contributions to the paper and provided assistance in data analysis and interpretation.

**Competing interests** The authors declare no competing interests.

**Additional information**

**Correspondence and requests for materials** should be addressed to Lamiya Mowla or Kartheik Iyer.

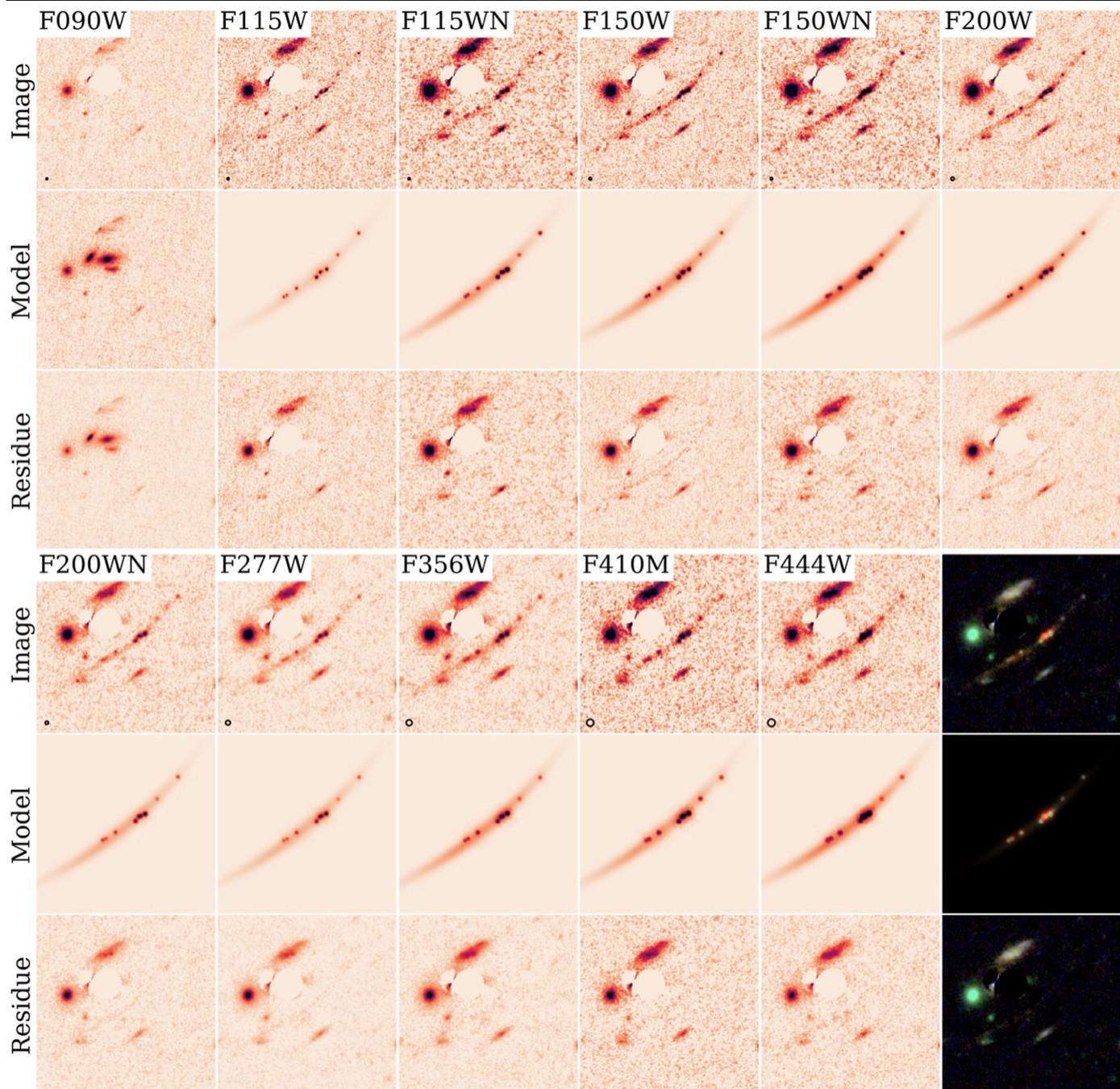

**Extended Data Fig. 1 | Morphological fit with GALFIT of the Firefly Sparkle in 11 JWST/NIRCam and NIRISS filters for photometric extraction and size determination.** The images, their respective models, and residues for 11 filters and the RGB image (R: F444W, G: F277W, B: 115W) are shown. The cutouts used for the fitting have size 10'×10'. In this figure we have zoomed in on the central 7'×7'. The FWHM of the point spread functions of the respective filters are shown as black circles on the lower left corner. The Firefly Sparkle is completely invisible in the bluest filter (F090W). Based on the reduced $\chi^2$ of the fits, nine out the ten clusters of the Firefly Sparkle are consistent with being point sources in F115W. The full model consists of nine point sources, an elliptical Gaussian for cluster (FF-4), and an elliptical Gaussian with a bending mode for the diffuse arc. The photometry is derived from the total model flux of the 11 components. The error of the photometry is estimated by injecting the full model in random locations in the MACS 1423 field, and refitting them them with GALFIT. The upper limit on the size of all clusters is determined by the HWHM of the F115W PSF (0.02'), as the deconvolved size of FF-4 is smaller than the F115W PSF.

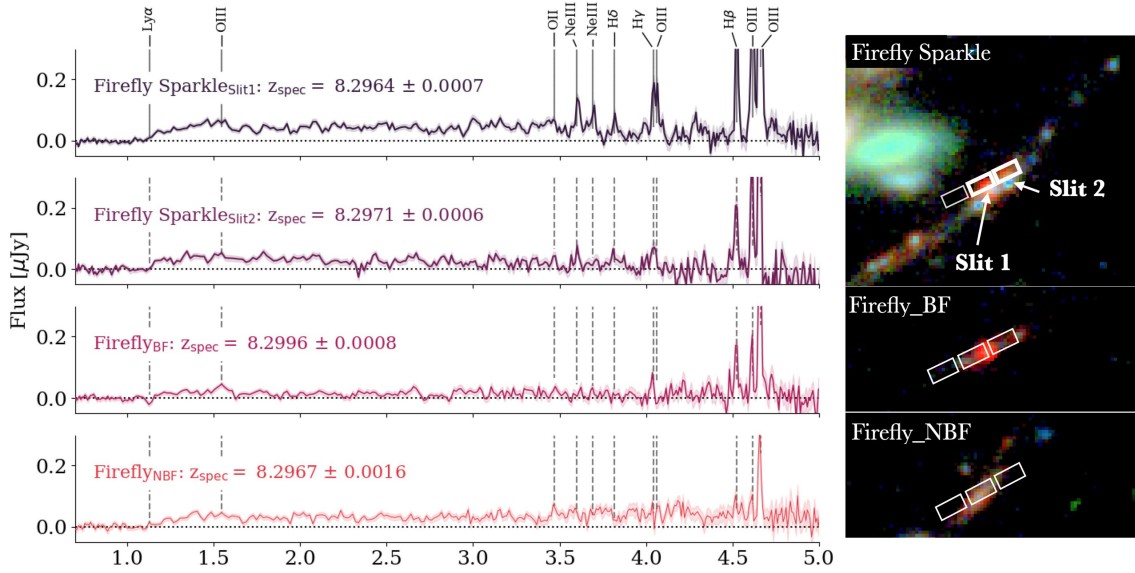

**Extended Data Fig. 2 | NIRSpec Prism spectra for two slits of the Firefly Sparkle, along with those of the nearby FF-BF and FF-NBF companions.** Slit 1 is dominated by FF-6 and contains contributions from FF-5 in Fig. 1, while Slit 2 is dominated by FF-4 and contains contributions from FF-3, FF-5 and FF-6, with both slits getting marginal contributions from the diffuse arc. Strong emission lines and a Lyman break in all the spectra unambiguously determine the redshifts of all the components. There is a slight oversubtraction of background at $\lambda > 4\,\mu$m for Firefly SparkleSlit 2 and BF due to their locations close to the bar of the NIRSpec MSA shutter. Further analysis of these regions is left for follow-up observations.

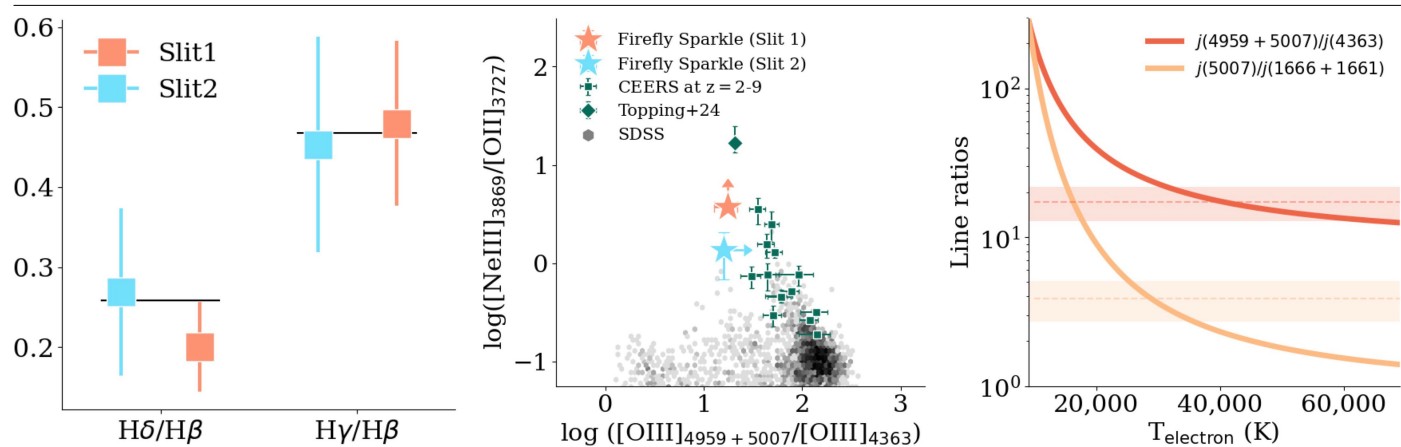

**Extended Data Fig. 3 | Inferring properties from the NIRSpec Prism spectrum of the Firefly Sparkle. Left:** Balmer decrements of Hδ/Hβ and Hγ/Hβ in Slit 1 (red) and Slit 2 (blue) spectra. Black solid line denotes the line ratios under Case B recombination. The line ratios indicates the dust attenuation is not significant in both spectra. **Middle:** Similar to bottom middle in Fig. 2, but Ne3O2 ratio is used instead as an indicator of the ionization parameter. **Right:** Electron temperature measurements from [O III] emission lines in Firefly Sparkle Slit 1. The dashed lines with shaded regions are the measured line ratios. The solid lines denotes predicted line ratios as a function of different electron temperatures from PyNeb[45]. The two different emission line ratios independently suggest a high electron temperature of $T_{e,O^{++}} \sim 40000$ K.

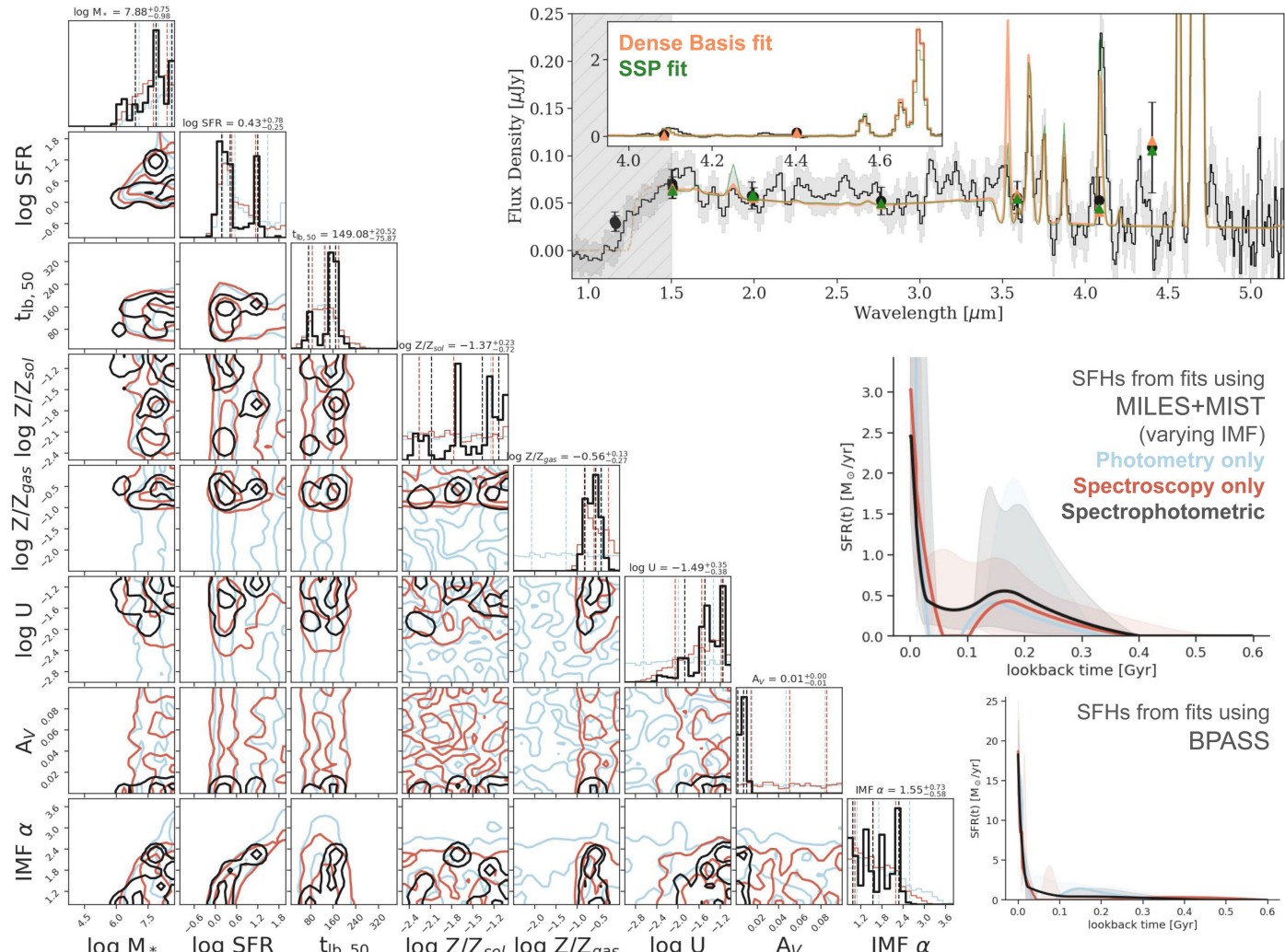

**Extended Data Fig. 4 | Inferring physical properties from the spectrophotometric fits.** The top panel shows the spectrophotometry for Slit 1 (black line and points) along with the best-fit spectrum from Dense Basis (orange line and points) and fits using simple stellar populations (green line and points). An inset panel shows the region with Hβ + [OIII] where the spectrum has much higher high fluxes. The corner plot shows the posteriors for each parameter being fit with only photometry (light blue contours), only spectroscopy (red contours) and both (black contours) using Dense Basis. The contours show the 1-σ and 2σ regions for each posterior, along with their covariances, while the diagonal plots show the marginal posteriors for each parameter. In addition to the joint posteriors, the spectra and photometry posteriors generally agree, with the spectra better able to constrain parameters like the gas-phase metallicity. The inset panels on the right show the stellar population posteriors from photometry alone (light blue), spectroscopy alone (red) and joint (black lines) along with 1σ uncertainties using the MILES+MIST and BPASS templates, again finding that they agree between photometric and spectroscopic fits within uncertainties.

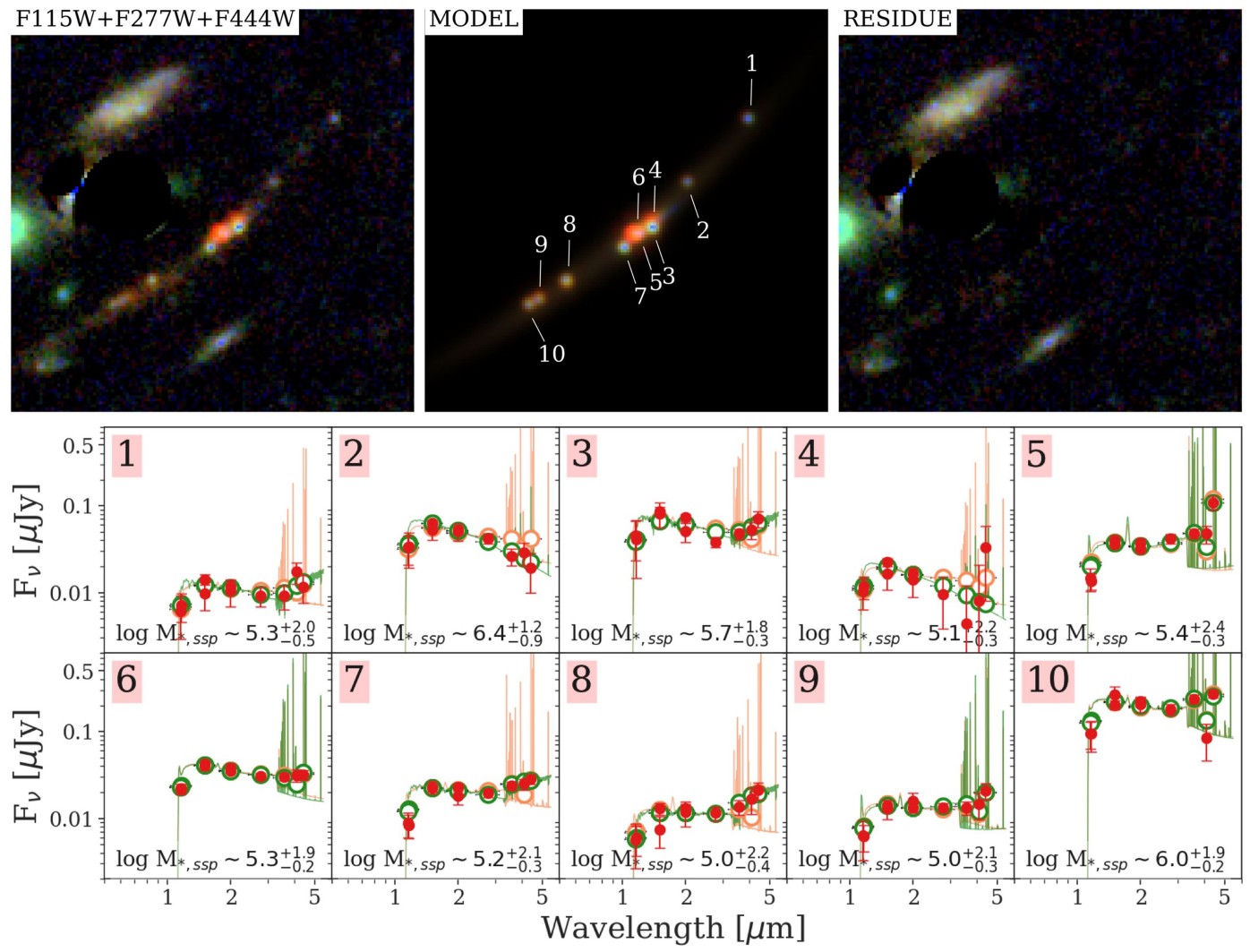

**Extended Data Fig. 5 | Top: The multiwavelength 11-component model for the resolved structure in the Firefly Sparkle consisting of 10 clusters and the diffuse arc shown in Extended Data Fig. 1.** The three panels show the observed image (left), the GALFIT model (middle) and the residuals (right) in composite F115W + F277W + F444W images. **Bottom:** Photometry for the 10 clusters are shown along with fits using DENSE BASIS (orange) and simple stellar populations (green), along with estimated stellar masses from the SSP fits.

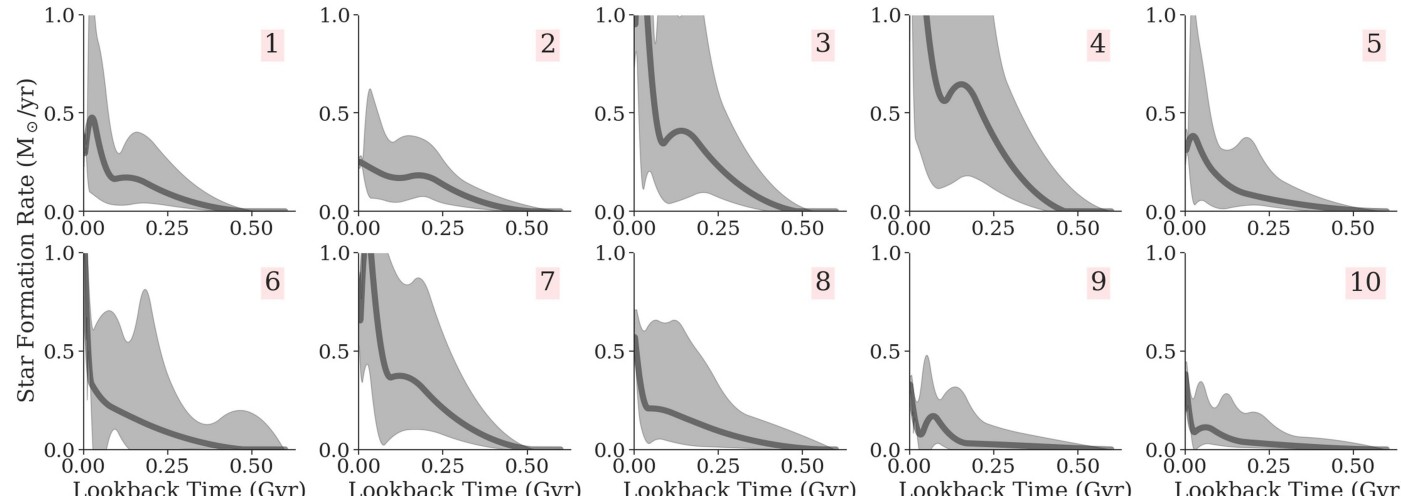

**Extended Data Fig. 6 | The SFHs of the ten star clusters from the DENSE BASIS-MILES+MIST fits.** The solid lines and shaded regions indicate the median and 1σ uncertainties for the SFHs of individual clusters.

**Extended Data Table 1 | Emission line flux measurements and the physical properties from the spectrum of Firefly Sparkle Slit 1**

| Line | Flux | Properties | values |
|------|------|-----------|--------|
| [O III]$\lambda$5007 | 646.8 $\pm$ 10.2 | $T_{e,\mathrm{O}^{++}}$ (from [O III]4363) | $4.0^{+2.6}_{-0.9} \times 10^4$ K |
| [O III]$\lambda$4959 | 202.0 $\pm$ 9.0 | $T_{e,\mathrm{O}^{++}}$ (from [O III]1666) | $2.9^{+0.7}_{-0.4} \times 10^4$ K |
| H$\beta$ | 125.1 $\pm$ 7.6 | $12 + \log(\mathrm{O}^{++}/\mathrm{H}^+)$ | $7.05^{+0.22}_{-0.37}$ |
| [O III]$\lambda$4363 | 47.5 $\pm$ 10.2 | $12 + \log(\mathrm{O}^+/\mathrm{H}^+)$ | $< 5.90$ |
| H$\gamma$ | 60.0 $\pm$ 12.4 | $12 + \log(\mathrm{O/H})$ | $7.05^{+0.22}_{-0.37}$ |
| H$\delta$ | 25.0 $\pm$ 6.8 | $\log(U_{\mathrm{ion,O32}})$ | $> -1.93$ |
| [Ne III]$\lambda$3968 | 43.0 $\pm$ 7.9 | $\log(U_{\mathrm{ion,Ne3O2}})$ | $> -1.96$ |
| [Ne III]$\lambda$3869 | 75.8 $\pm$ 8.5 | | |
| [O II]$\lambda$3727 | <20.2 | | |
| [O III]$\lambda\lambda$1661,1666 | 166.5 $\pm$ 50.7 | | |

Fluxes are in units of $10^{-20}$ erg s$^{-1}$cm$^{-2}$.

**Extended Data Table 2 | Median stellar mass and age estimates from the various SED modeling configurations described in Section 4 in Methods**

| Fit type Models | $\log M_{*,50}$ (SSP) (MIST) | $\log M_{*,50}$ (DB) (MIST) | $\log M_{*,50}$ (SSP) (BPASS) | $\log M_{*,50}$ (DB) (BPASS) | $t_{age}$[Myr] (SSP) (MIST) | $t_{50}$[Myr] (DB) (MIST) | $t_{age}$[Myr] (SSP) (BPASS) | $t_{50}$[Myr] (DB) (BPASS) | $\chi^2$ (SSP) (MIST) | $\chi^2$ (DB) (MIST) | $\chi^2$ (SSP) (BPASS) | $\chi^2$ (DB) (BPASS) |
|---|---|---|---|---|---|---|---|---|---|---|---|---|
| FF-1 | 5.26 | 5.91 | 6.28 | 6.67 | 5.27 | 96.84 | 4.56 | 108.95 | 0.23 | 0.10 | 0.23 | 0.26 |
| FF-2 | 6.36 | 5.71 | 6.87 | 7.05 | 8.44 | 151.32 | 5.28 | 36.32 | 0.18 | 0.37 | 0.89 | 2.31 |
| FF-3 | 5.66 | 6.26 | 6.87 | 7.16 | 3.83 | 90.79 | 4.01 | 36.32 | 1.21 | 1.50 | 1.00 | 1.31 |
| FF-4 | 5.14 | 6.35 | 6.29 | 6.67 | 4.23 | 115.00 | 3.83 | 12.11 | 0.30 | 1.73 | 0.35 | 0.44 |
| FF-5 | 5.41 | 5.74 | 6.60 | 7.05 | 3.64 | 84.74 | 3.91 | 163.42 | 0.44 | 0.47 | 0.53 | 0.56 |
| FF-6 | 5.31 | 6.04 | 6.67 | 6.84 | 2.91 | 84.74 | 6.29 | 42.37 | 0.26 | 0.61 | 0.09 | 0.23 |
| FF-7 | 5.15 | 6.04 | 6.34 | 6.77 | 4.14 | 96.84 | 4.27 | 133.16 | 0.14 | 0.25 | 0.12 | 0.12 |
| FF-8 | 5.01 | 5.80 | 6.09 | 6.48 | 4.94 | 115.00 | 4.41 | 169.48 | 0.25 | 0.37 | 0.28 | 0.28 |
| FF-9 | 4.97 | 5.57 | 6.13 | 6.55 | 4.05 | 84.74 | 3.59 | 60.53 | 0.10 | 0.37 | 0.12 | 0.10 |
| FF-10 | 6.00 | 5.64 | 7.27 | 7.63 | 2.13 | 84.74 | 2.17 | 6.05 | 0.49 | 0.10 | 0.42 | 0.68 |
| FF-arc | 5.43 | 6.72 | 6.83 | 6.44 | 3.56 | 90.79 | 0.28 | 12.11 | 14.91 | 0.53 | 21.25 | 44.75 |
| FF | - | - | - | - | 0.80 | 36.32 | 0.67 | 12.11 | 3.20 | 18.06 | 1.54 | 1.71 |
| FF-NBF | - | - | - | - | 3.61 | 121.05 | 232.01 | 12.11 | 21.28 | 4.17 | 41.72 | 48.00 |
| FF-BF | - | - | - | - | 33.77 | 84.74 | 44.93 | 169.48 | 0.53 | 0.23 | 1.08 | 0.82 |

**Extended Data Table 3 | Magnification and upper limit on the size of the Firefly Sparklestar clusters, the diffuse arc, BF, and NBF**

| ID | $\mu$ ($\mu_{tan}$) | $r_{50}$ (pc) |
|---|---|---|
| FF-1 | $15.6^{+3.1}_{-2.0}$ ($13.9^{+0.9}_{-2.8}$) | $< 6.8^{+0.4}_{-1.3}$ |
| FF-2 | $17.1^{+3.7}_{-2.2}$ ($15.2^{+1.0}_{-3.2}$) | $< 6.2^{+0.4}_{-1.3}$ |
| FF-3 | $20.6^{+4.6}_{-3.4}$ ($18.6^{+1.5}_{-4.3}$) | $< 5.1^{+0.4}_{-1.2}$ |
| FF-4 | $21.1^{+4.8}_{-3.5}$ ($19.0^{+1.6}_{-4.5}$) | $< 4.9^{+0.4}_{-1.2}$ |
| FF-5 | $22.5^{+5.4}_{-3.9}$ ($20.5^{+1.8}_{-5.0}$) | $< 4.6^{+0.4}_{-1.1}$ |
| FF-6 | $23.7^{+6.0}_{-4.3}$ ($21.8^{+2.0}_{-5.5}$) | $< 4.3^{+0.4}_{-1.1}$ |
| FF-7 | $24.7^{+6.5}_{-4.6}$ ($23.0^{+2.3}_{-5.9}$) | $< 4.1^{+0.4}_{-1.1}$ |
| FF-8 | $26.1^{+8.0}_{-4.6}$ ($24.3^{+3.0}_{-6.3}$) | $< 3.9^{+0.5}_{-1.0}$ |
| FF-9 | $24.2^{+7.5}_{-3.9}$ ($22.2^{+2.9}_{-5.4}$) | $< 4.2^{+0.6}_{-1.0}$ |
| FF-10 | $24.0^{+7.4}_{-3.9}$ ($22.0^{+3.0}_{-5.3}$) | $< 4.3^{+0.6}_{-1.0}$ |
| FF-arc | $24.4^{+6.0}_{-6.0}$ | - |
| Firefly Sparkle | $24.4^{+6.0}_{-6.0}$ | - |
| BF | $28.0^{+14.4}_{-3.6}$ | - |
| NBF | $9.1^{+1.0}_{-0.9}$ | - |

**Extended Data Table 4 | Photometry of individual star clusters and the diffuse arc of the Firefly Sparkle**

| ID | F115W | F115WN | F150W | F150WN | F200W | F200WN | F277W | F356W | F410M | F444W |
|---|---|---|---|---|---|---|---|---|---|---|
| FF-1 | $15.6 \pm 1.6$ | $9.3 \pm 2.1$ | $19.2 \pm 1.1$ | $13.6 \pm 1.0$ | $16.8 \pm 1.5$ | $14.4 \pm 1.8$ | $11.1 \pm 1.4$ | $10.3 \pm 1.0$ | $10.5 \pm 1.3$ | $11.8 \pm 1.9$ |
| FF-2 | $9.8 \pm 1.5$ | $2.1 \pm 1.2$ | $13.6 \pm 1.1$ | $4.6 \pm 1.8$ | $9.1 \pm 0.9$ | $6.8 \pm 2.1$ | $5.2 \pm 1.2$ | $6.1 \pm 1.4$ | $13.6 \pm 2.2$ | $8.0 \pm 2.6$ |
| FF-3 | $39.6 \pm 5.6$ | $26.2 \pm 3.7$ | $65.0 \pm 1.6$ | $39.6 \pm 1.3$ | $50.0 \pm 1.3$ | $37.5 \pm 1.1$ | $34.0 \pm 4.3$ | $24.9 \pm 1.3$ | $24.4 \pm 2.4$ | $23.7 \pm 3.0$ |
| FF-4 | $15.4 \pm 10.2$ | $15.2 \pm 7.4$ | $42.3 \pm 2.2$ | $45.0 \pm 0.8$ | $37.3 \pm 1.3$ | $39.0 \pm 1.8$ | $27.9 \pm 8.3$ | $27.3 \pm 2.2$ | $32.2 \pm 3.8$ | $74.9 \pm 4.8$ |
| FF-5 | $7.5 \pm 4.0$ | $10.6 \pm 3.6$ | $14.9 \pm 1.4$ | $21.9 \pm 2.7$ | $22.7 \pm 1.8$ | $18.0 \pm 2.3$ | $16.2 \pm 1.9$ | $18.1 \pm 2.2$ | $18.2 \pm 4.0$ | $30.2 \pm 4.1$ |
| FF-6 | $31.3 \pm 3.7$ | $19.8 \pm 3.0$ | $58.0 \pm 1.6$ | $46.2 \pm 3.4$ | $37.1 \pm 1.5$ | $39.2 \pm 1.6$ | $42.6 \pm 1.9$ | $48.2 \pm 3.1$ | $49.9 \pm 4.3$ | $109.6 \pm 4.0$ |
| FF-7 | $28.4 \pm 1.8$ | $23.0 \pm 1.9$ | $50.6 \pm 0.6$ | $42.1 \pm 1.8$ | $39.0 \pm 1.1$ | $40.4 \pm 1.8$ | $30.6 \pm 1.0$ | $29.7 \pm 1.1$ | $29.4 \pm 2.1$ | $32.4 \pm 2.3$ |
| FF-8 | $11.2 \pm 1.5$ | $7.0 \pm 1.1$ | $23.1 \pm 1.5$ | $21.3 \pm 1.3$ | $20.1 \pm 1.3$ | $16.7 \pm 1.7$ | $18.0 \pm 0.5$ | $22.5 \pm 0.8$ | $21.1 \pm 1.4$ | $26.9 \pm 1.5$ |
| FF-9 | $7.5 \pm 1.9$ | $4.0 \pm 1.0$ | $15.0 \pm 1.1$ | $8.1 \pm 0.8$ | $13.4 \pm 0.9$ | $11.2 \pm 1.8$ | $11.0 \pm 0.8$ | $12.9 \pm 0.9$ | $13.2 \pm 2.0$ | $17.1 \pm 1.6$ |
| FF-10 | $8.0 \pm 2.2$ | $6.8 \pm 0.8$ | $16.4 \pm 1.3$ | $11.3 \pm 1.8$ | $14.8 \pm 1.3$ | $15.1 \pm 1.3$ | $12.1 \pm 1.0$ | $12.5 \pm 1.1$ | $10.9 \pm 1.8$ | $17.7 \pm 2.3$ |
| FF-arc | $94.6 \pm 20.0$ | $168.9 \pm 7.2$ | $290.2 \pm 16.8$ | $355.4 \pm 4.3$ | $329.7 \pm 8.6$ | $331.4 \pm 13.3$ | $259.6 \pm 8.6$ | $308.5 \pm 8.9$ | $190.2 \pm 8.8$ | $372.4 \pm 5.5$ |

Fluxes are in units of nJy.