## [Peer Review File · Nature]

Manuscript Title: Formation of a low-mass galaxy from star clusters in a 600 Myr old Universe

Reviewer Comments & Author Rebuttals

Reviewer Reports on the Initial Version:

Referee #1 (Remarks to the Author):

This paper presents new observations and an analysis of a lensed galaxy at high-redshift, detailing the physical properties of a possible Milky Way progenitor in the early Universe. The high-quality JWST imaging and spectroscopy alone provides tremendous value and would be worthy of publication on its own. I find the authors' analysis of the data and its implications for our understanding of galaxy formation to be thorough and robust. I have some minor comments and suggestions to make to improve the paper, which I detail below. Once those are addressed, I would recommend this paper for publication.

One general comment I have is that in several places throughout the paper, the magnification or magnification range is stated. Other quantities that depend on the magnification, such as the size, stellar mass, etc., are also presented in many places. These should either always include the magnification uncertainties due to uncertainty in the lens model, or it should be stated clearly the first time the magnification is mentioned that the quantities either do or do not include them. It is not always clear whether some of the physical quantities presented in the paper factor in these uncertainties in the magnification.

Figure 2, top right panel: Are Slit 1 and Slit 2 mislabeled in the 2D spectrum? The top trace (currently labeled "Slit 2") shows emission lines that appear in the 1D spectrum, even though the 1D spectrum is supposed to be Slit 1.

Figure 2, bottom left panel: It's hard to evaluate this χ^2 map and what region of parameter space is allowed by the fit. Is it possible to make this a contour plot showing 68% and 95% quantile contours instead?

Figure 2, bottom right panel: Is this plot for Slit 1 and Slit 2 combined?

Figure 3, top right panel (and other similar panels throughout the paper): The x-axis is a bit unclear. Is time in the +x direction actually time prior to the source redshift? The axis label and/or caption should more clearly reflect this, or add a second x-axis showing redshift.

Figure 3, bottom panels: The grey arrows seem unnecessary since the plots are clearly numbered. The arrow from 6 doesn't seem to be pointing to the correct object.

Figure 4, right panel: It seems like the x-axis range could be made smaller so we can see the distribution of the clusters better. Is there some easy way to indicate which clusters are which on this plot?

Figure 5: Indicate the size of the cutout in the caption or on the image itself. Does the circle in the bottom left of the images represent the PSF size? This should be stated in the caption.

Section 1.2, 2nd paragraph: When fitting the light components, are any of the parameters linked across the various filters, or are they all fit independently?

Section 1.2, 3rd paragraph: "...we inject the full Firefly Sparkle model in 100 random locations in our 10"x10" postage stamps..." - do you mean that you inject the model in random locations in the entire field, not the postage stamp? How large is the entire field?

Section 1.3.1: When deriving the total oxygen abundance of $12+\log(\text{O}/\text{H}) = 6.99+0.15/-0.33$, does this account for the different $T_{e,O}$ measurements and their uncertainties? What about the assumption of n_e ?

Section 1.3.1: "Both ratios provide similar limit of $\log(U) > -2.1$ " - you haven't defined U yet.

Section 1.4, last paragraph: In determining the properties of each cluster, how much do the results change if you don't fix the mass slope or change the assumed value to something else? Why do you assume $\alpha = 1.0$ instead of the central value of $\alpha = 1.3$ from the fit to the data?

Table 2: Related to the previous point, it should be stated in the caption what assumptions are made here, and what the uncertainties actually reflect. The error bars seem quite small if all uncertainties (magnification, IMF, etc.) are taken into account.

Table 2: Why isn't there a magnification for the FF-arc? While the magnification is different at various locations along the arc, you can presumably calculate a total flux to de-lensed flux to get the magnification. Is this how the other quantities for the FF-arc are derived? The same can apply to BF and NBF.

Section 1.5, 1st paragraph: Were there any new multiply-imaged systems discovered in the JWST photometry? If so, why were they not used for the modeling? Have there been any independent mass models of this system?

Section 1.7: In the main paper text, it is stated that the stellar mass of the Firefly Sparkle is $\sim 10^{6.8} M_{\text{sun}}$ and that it is consistent with the stellar masses expected for a MW or M31 progenitor. This section describes the procedure, but doesn't give any values for the range of stellar masses derived for such progenitor. This range should be given here for comparison, and to give a sense of the uncertainties in the procedure.

Referee #2 (Remarks to the Author):

I have read the paper “The Firefly Sparkle: The Earliest Stages of the Assembly of a Milky Way-type Galaxy in a 600 Myr Old Universe” with interest. This paper presents a joint photometric and spectroscopic study of a gravitationally lensed arc at $z=8.3$, which exhibits a clumpy structure. The clumps sizes and stellar masses are consistent with early proto-globular clusters, and the spectra show nebular features that suggest a top-heavy initial mass function. The authors conclude that this galaxy is a likely progenitor for Milky-Way like galaxies. These results are compelling and warrant publication, however several issues will need to be addressed before I consider this manuscript to be ready for publication. I list major and minor points below.

Major:

- The description of the NIRSpec data reduction is insufficient as currently presented. NIRSpec prism data is background limited, and the continuum shape is used in later analysis, so proper background subtraction is critical to ensure that the final result is not being influenced by leftover sky background. In its current state, the paper cites Asada et al. in prep. for the custom background subtraction method. With the Asada paper not available, it is not possible to assess the reliability of background subtraction, and thus not possible to assess the reliability of any of the continuum modeling described in the subsequent section. The authors either need to make the Asada paper public (an arxiv posting is sufficient) or they need to include the relevant details of the custom background subtraction in this paper.
- I am somewhat concerned about the [OIII]1660,6 and [OIII]4363 detections, and thus the validity of the unusually high electron temperature measurements. Based on the line flux uncertainties quoted in Table 1, each of these are detected at $\sim 4\sigma$. However, looking at the spectrum in Figure 2, I see several features that are not fit as emission/absorption lines that appear (by eye) to have similar SNR. Could the authors please define the criteria by which emission features were selected? Related, it has been found that the NIRSpec reduction pipeline underestimates the uncertainty of the final spectra (see e.g. Appendix B, <https://ui.adsabs.harvard.edu/abs/2023ApJ...951L..22A/abstract>) . It is not clear how the uncertainty spectra are calculated with msaexp. Could the authors please clarify how the uncertainty spectra are calculated, and whether or not they are similarly underestimated?
- The text states that FF-4 “exhibits an elongated component”, and is thus fit with a Gaussian ellipse. However, the major axis is later stated to be smaller than the PSF, indicating that the object is unresolved. These two statements appear to be at odds with each other - an unresolved clump should look like the PSF, and thus should not appear elongated. Looking at Figure 5, it appears to me that the clump is indeed unresolved. Could the authors please either (1) edit the text to consistently refer to this clump as unresolved, or (2) provide additional details (e.g. from the GALFIT modeling) to show that the clump is partially resolved?
- The authors state in Methods Section 1.2, “The agreement between NIRISS and NIRCcam fluxes in the three overlapping filters is another confirmation of the robustness of photometry”. However, looking at Table 3 and the bottom panel of Figure 9, it seems like many of the overlapping filters are in significant tension with each other (some at the 3 sigma level or greater). This would seem to contradict the statement in the text. Could the authors please explain this tension?

- Could the authors please elaborate on their reasoning for fixing certain parameters in the photometric SED fits for all star clusters based on the spectrophotometric results that cover only a subset of the clusters? It is stated that clusters 3-6 are covered with the spectroscopic observations. Based on the source plane model shown in Fig. 3, cluster 7 appears close enough to be associated with clusters 3-6, but clusters 1,2, and 8-10 seem to be far enough away that it is not clear they will have consistent properties.

Minor:

- Main text, paragraph 2 - please specify how the range of magnifications is determined (based on table 2, it seems this is a range of best fit magnifications, but does not account for uncertainties?)
- Fig 2 - To my eye, the emission lines in Slit 2 in the 2D spectrum appear brighter than those in Slit 1, however your extracted 1D spectra show the opposite. Could you more clearly delineate the 2D spectra of Slit 1 and Slit 2? I trust your 1D extractions, I'm just struggling to interpret the figure.
- Page 7, first full paragraph - "...only spectrum with SNR high enough to constrain the slope" - I'm guessing "the slope" here refers to the IMF slope? Please clarify this in the text.
- Page 8 - "given the extreme shear...it is unlikely that the magnification will significantly change" - This seems backwards to me. Typically objects with high magnifications have high uncertainties, yet this sentence appears to indicate the opposite. Do you mean that the large shear implies the magnification will remain high as the lens model is refined?
- Methods Sec. 1.2, Paragraph 2 - Could the authors please elaborate on their construction of an empirical PSF? Which stars are used/what criteria are used to select them?
- Methods Sec. 1.2, end of Paragraph 2 - "Residuals from the fit are negligible" - By what metric are the residuals determined to be negligible?
- Methods Sec. 1.2, Paragraph 3 - "we find no significant systematic offset..." Please quantify the offset, and state which metric was used to determine its insignificance.
- Section 1.3 - Were all objects (FF, BF, NBF) observed in all 3 MSA configurations? And thus was each object observed with the same exposure time?
- Fig. 7 - The caption here does not provide much information about the figure. Please expand this figure caption.
- Page 15 - please cite the solar abundance determination used.
- Sec 1.4, first paragraph - the prior on IMF high-mass slope appears incomplete?
- Sec. 1.4, last paragraph - Please specify which IMF high-mass slope value is used for the photometric fitting. The current text ($\alpha \sim 1$) is not clear.
- Sec. 1.5 - If the lens modeling paper (Desprez et al.) is not publicly available, please include a figure showing lens model constraints and the lensing critical curve.

Referee #3 (Remarks to the Author):

Dear authors,

I have read with great interest the manuscript “The firefly Sparkler: The earliest stages of the Assembly of a Milky Way-type galaxy in a 600 Myr old Universe”. The manuscript presents the spectra-photometric analysis of a remarkable lensed galaxy at spectroscopic redshift of 8.3. The Firefly sparkler (FF) is an interacting system with at least 2 more galaxies located at similar redshift. I have several major concerns that affect many of the key conclusions the authors reach. In many points, the manuscript lacks clarity or connection between the different topics. Important aspects connected to the conclusions drawn in the manuscript are in the methods, while text that is not relevant is included in the main text. Many statements are made and not refereed. Many assumptions made in the analysis but are incorrect and affect the entire analysis and interpretation of the results.

1. It is difficult for the reader to read through the main story the article is aiming at. The summary and main text of the article jumps from one topic (star clusters) to the other (ionisation status of the ISM of the galaxy), with a third topic (potential interaction with other galaxies) making the appearance, but there is no clear connection between the three topics in the text. In my humble opinion, the connection throughout the text should be the star clusters. The authors have the unique, truly unique, opportunity to link cluster feedback at physical scales of a few parsecs to the extreme ISM conditions that JWST is revealing in many of the reionization era galaxies. I believe that that is the real breakthrough of this object and work. However, the authors dwell into globular cluster formation and Milky Way progenitors, which I actually find weak and circumstantial at best (a lot of speculation and little results). The authors misuse SED fitting techniques to fit star clusters. A galaxy can have a non-parametric SFH, a star cluster in local or high redshift universe is a stellar system of a few parsec scales that form in an instantaneous event. We have 30 years of literature to support that, both from numerical and observational constraints, and this cannot simply be ignored. The authors at some point mention in the text that only non-parametric SFHs can reproduce elevated EWs. Well, this might be true for galaxies, but actually simple stellar populations produce normally (even in the local universe) EW of 1000 Å. Therefore, I urge the authors to use the information at hand, i.e., detecting bound star clusters to actually use SFH that assume a delta burst. This can be done with a short exponential tau of 1 or 10 Myr, a burst assumption, a SSP. Moreover, it is very important to not assume an underlying continuous star formation history. The way photometry has been performed ensures that the diffuse galaxy light has been removed from the cluster light. Once the authors make this change in the assumed SFH to fit the cluster light they will find extremely young ages and finally quantify the amount of feedback that each of these little sparks are producing. I expect that the cluster producing the ionisation detected in slit1 is about 3 Myr old. The other clusters will show age gradient moving away from this central cluster. I would encourage the authors to include citations to work done at lower redshift where cluster feedback has been evaluated. They should include citations to Toppings et al 2024, Vanzella et al 2023 (Sunrise arc), Vanzella et al 2023 (Lap1), any of the many works done in the sunburst arc, the first confirmed star cluster leaker at redshift 2.4.

2. Regarding all the circumstantial results collected to probe the IMF shape. None of them reads convincingly. I bring up the work done by Cameron et al 2024, which has been widely referred to in the manuscript. Cameron and collaborators find indeed galaxies that show a Balmer jump and a steep turnover at $<1430 \text{ \AA}$. Both features point toward an elevated electron temperature as well as the need of a very hot stellar spectrum, which combined with the nebular continuum can explain the strengths of the turnover in the FUV.

I. Differently to what done by Cameron and collaborator, the authors do not include any treatment of IGM absorption (or a DLA in front of the galaxy). There is no mention of it throughout the analysis description. I don't need to explain that the light of a galaxy at redshift 8.4 will still be affected by neutral hydrogen absorption, either in proximity of the targets or along the line of sight. That such absorption is playing a role could be seen in the absence of Ly α emission (even if the latter could be missing because of the prism resolution, although notice the booming optical emission lines...). It is very important that the authors produce clear evidence of a turnover due to extreme ISM and ionisation sources after IGM and DLA absorptions have been taken into account.

II. The second necessary step before claiming top-heavy IMFs is to use correct stellar population models in the SEDs. Again, the authors appear to ignore that in star clusters the most massive stars that can be produced reach easily 300 Msun, moreover stripped stars could provide a hot ionising source. The authors have so far assumed a typical Kroupa IMF. They do not even specify what upper-mass limit cutoff they use in the fit. Hence, I would like to see fits where the IGM absorption is accounted for and normal IMF with higher mass cutoff and binary populations (like those produced in BPASS) are used. It remains unclear and it is not explained in the text why the authors fit all the spectrum, but then when they fit photometry and spectrum combined, they stop well before the IGM absorption part (and exclude the F115W). This is not explained in the text although it is done in the analysis presented in Fig 7 and 9. The observed SED is not done on 10 bands but effectively on 6 since the NIRISS data do not provide any further constraint than NIRCcam filter and the bluest filter is removed.

III. Since assumptions from the spectrophotometric analysis are propagated to the fit of the clusters, the latter analysis does not provide any constraint on the IMF shape.

3. Abundance matching method to link the FF to Milky Way assembly. This method is based in many assumptions and extrapolations. The authors do not have dark matter halo mass at hand and prescription to associate mass at this redshift are truly unreliable. Looking at the broad work presented in the literature that has tried to simulate star cluster formation and evolution during galaxy assembly (see, Grudic et al 2023, and all references therein for the very large literature!!) there is agreement to the fact that the Milky Way globular clusters with masses above 10^5 Msun start to form at significantly lower redshift ($z < 7$) than the FF galaxy. If we rely on works that actually look at Milky Way-like disks, there is no evidence that the FF galaxy will be a progenitor of the MW. More likely it will be a progenitor of an early galaxy. In the text the authors write "We find that the Firefly Sparkle has a total demagnified stellar mass of $\log M \sim 6.8 \text{ Msun}$ ". Does this estimate include the mass in star clusters? It is misleading since in the text the authors quote only the diffuse light mass, but the clusters are an important component in such young galaxy and cannot be neglected. Table 2 report some masses but not the magnification value used to derive the lensing-corrected mass. Please provide in the text the total mass of the galaxy and in Table 2 an entry for diffuse component and for total galaxy light (cluster + diffuse).

I will now follow the organisation of the text:

1. Summary. There are several statements which are not supported by the literature. For example: “Composed today primarily of old stars and correlating with the properties of their parent dark matter halos, the first globular clusters are thought to have formed during the earliest stages of galaxy assembly.” Actually, this simplified idea is not that obvious and actually does not reflect what is known in the GC community. An idea of where we stand with understanding of GC formation in the Milky Way progenitor can be grabbed from the following work and reference therein. The lack of GCs with metallicities below the -2.5 floor suggests that the GCs in the Milky Way formed at $z < 6$.
<https://ui.adsabs.harvard.edu/abs/2019MNRAS.486L..20K/abstract>

2. Summary. “The mass distribution of the galaxy seems to be concentrated in ten distinct clusters ($\sim 49 - 57\%$ of the total mass), with individual cluster masses ($M_{*,\text{cluster}} \sim 105.3 - 105.8 M_{\odot}$) that straddle the boundary between low-mass galaxies and high-mass globular clusters.”
I’m not sure why is relevant to quote in the summary the ultra-faint dwarfs? Ultra-faint dwarfs have dark matter the star clusters that the authors are studying do not. I would rather find more significant/relevant a reference to the peak of the GC max function which is $\log(M) \sim 5.2 M_{\text{sun}}$ which is close to be constant in all GC populations studied at redshift 0.

3. Summary. “The cluster ages suggest that they are gravitationally bound with star formation histories showing a recent starburst possibly triggered by the interaction with a companion galaxy at the same redshift at a projected distance of ~ 2 kpc away from the Firefly Sparkle.” How can single stellar populations show star formation history in their interiors that can tell a recent interaction has taken place in the galaxy? The age of the star clusters should be referred to. Unfortunately, with the current assumptions made in the analysis cluster ages cannot be used to trace the recent burst history of the galaxy and the propagation of star formation within the galaxy.

4. “Due to its high [O iii] equivalent width (EW) contributing to the F444W broadband flux, this source can appear to be a double-break source similar to those in Labbe et al. 6 and Desprez et al. 7”. I believe that this is redundant to the context of the manuscript. Please remove.

5. GALFIT analysis. From text in main manuscript and text in the Method section is not clear how the photometry is derived. Do the authors allow the shapes of clusters and diffuse light to change from filter to filter or is fixed to the ones extracted in the F444W or F150W? I’m worried that if the authors let the shape to be fit in each band and allow all parameters free, they all be dominated by differences in the faint low S/N regions and therefore they include in each band different physical areas in the different components (clusters, galaxy light). This will result in differences in the recovered SED shape, thus affecting the reliability of all derived parameters. This is mostly concerning since by looking at the SEDs in figure 9. The flux in the same filter but different instruments produce significantly different fluxes. Yet there is no mention in the text about this problem or clear explanation about the way the photometry is extracted. Finally, I would like to stress that looking at the residuals in figure 5 significant light remains at the location of the clusters and visible in all the band separately and showing up in the color composite at the bottom-right.

6. I find this statement redundant “Resolved photometry is also necessary for estimating the total mass of the Firefly Sparkle, as global spectral energy distribution (SED) fitting can bias stellar masses when a young stellar population outshines the first episodes of star formation (e.g., 9–11)”. It seems from Table 32 and comments in the text that the fit to the diffuse light and that of the star clusters provide exactly the same output. On this point. It is not clear in Table 2 what values are reported. The star clusters have values that are corrected for magnification. While the galaxies do not have any magnification value quoted and therefore it is not clear if the mass reported for the 3 galaxies is corrected for magnification. Moreover, please notice that the total mass of the galaxy is the diffuse light+star clusters so I hope (but it is not clear from the text and tables) that that is what the authors quote for the galaxy mass. I think that it would be beneficial that the authors fit the total light of the galaxy+clusters to check whether they will recover similar results for the galaxy mass and SFR, SSFR.

7. Figure 2. In the top right panel, the authors write slit1 and slit2 but I believe it is inverted as from the text slit1 has the strongest lines that in the figure are labelled slit2.

8. “These features are quite similar to those observed in [19–23], suggesting that the nebular continuum is a significant contributor to the overall SED for this star-forming cluster.” The cluster cannot be star forming if its age is larger than 100 Myr (e.g., please fit with the correct SFH assumptions). Moreover, as the authors notice later on, the light of multiple clusters enters this region, therefore I would refrain from calling the source within slit 1 cluster. I would encourage the authors to use the term stellar clump, ensemble of stellar clusters and stellar light in a region that is compact but not exactly coinciding with a single cluster.

9. I cannot avoid to point out that I do not understand why the authors add 1 reference to Adamo et al 2024 work and only when discussing stellar densities. I believe that the comparison with that work is fundamental to understand similarities and differences. This manuscript presents the second highest evidence of star cluster formation in reionization era. The text of the manuscript touches on many points which are similarly discussed in Adamo et al., yet no reference is made to that work. Moreover, the concept of bound star clusters at high redshift has been proposed in the literature by Vanzella et al 2019-2020, then discussed in Metric et al 2022, Claeysens et al 2023 (the latter work also discusses the sparkler). No mention of any of the many works at high redshift that look for bound star clusters can be found in the text.

10. “Varying the power-law slope of the Kroupa IMF in FSPS results in an excess of high mass stars, which (i) increase the nebular continuum from the two-photon component [and therefore the strength of the Balmer jump; 19], and (ii) increases the equivalent width of the emission lines.” Single stellar populations can easily explain $\log(EW) \sim 1000$, no need to advocate top-heavy IMFs.

11. “The motivation for the Firefly Sparkle came from the Sparkler [29] - a strongly lensed galaxy at $z = 1.38$ surrounded by old star clusters that could be resolved only with JWST. In contrast to the Sparkler, the Firefly Sparkle represents one of JWST’s first spectrophotometric observations of an extremely lensed galaxy assembling at high redshifts, with clusters that are in the process of formation instead of seen at later epochs.”. Honestly, looking at the currently very old ages of the star clusters in the FF galaxy there is not really support to the claim of being clusters in the process of forming.

12. Section 1.6 of Methods. I have to honestly say that I do not understand what is written in this section:

I. The authors use the main text to introduce all quantities that will be used to produce Figure 4 of the main text. Yet all the discussion of that figure and implications are done in Section 1.6 that reads like a few paragraphs extracted from the main text of a previous version with more figures and different analyses presented here.

II. Why the authors show in Figure 3 and mention here a mass map? How is this mass map derived? Did the authors perform a px-to-px fit? There is no mention in the text. Why is this mass map relevant for the analysis? The authors mention the fit to the diffuse light of the galaxy and I understood from the text that is done extracting integrated fluxes.

III. "We approximate an upper limit on the 3D half-mass radius from our projected upper limit on half-light size ($r_h = 4/3r_{50}$). The density is calculated using $\rho \equiv 3M/(8\pi r_h^3)$, where M is the total demagnified mass of the star clusters and the diffuse arc (assuming $\mu = 25$)." Why is the diffuse arc mentioned here??

IV. "The resulting density vs. stellar mass plot is shown in left panel of Figure 4. "No such plot is presented in the paper.

V. "Here, the half-mass radius of the galaxy R is calculated from the mass map (shown in Figure 3." → how is the radius of the galaxy relevant to the estimate of the cluster density? Again, this is the mass map showed in Figure 3 and never introduced in the main text.

VI. The content of Figure 4 right is described as "we solve for Equation B6 in Gieles et al. [17], assuming $\zeta = 2$ for the early universe, and scaling the Milky Way rotation velocity to the mass of our galaxy." How can the authors justify this for an early galaxy at redshift 8??? There is no way to possibly justify that this galaxy is a scaled version of the Milky Way rotation. Dwarf galaxies have very little rotation and almost no shear. The model presented by Gieles et al is exclusively developed for spiral galaxies and cannot be simply scaled down to dwarf galaxies at redshift 8. The dynamics and kinematics of these early galaxies are unknown. Even in the local universe, the Gieles et al relation would not be applicable to dwarf galaxies.

VII. "This figure is showing us that most of these star clusters are expected to survive to the present-day universe, and will expand and then get ripped apart to form the stellar disk and the halo of the galaxy. The only way they survive is to get kicked out to large distances, away from the dense tidal field of the galaxy." while in the caption of Figure 3 it reads: "Majority of these star clusters are not expected to survive to the present, but will instead expand and tidally stripped apart to form the stellar disk and the halo. However, star clusters kicked out to large galactocentric radii may survive to become present day globular clusters." So, what is the conclusion the reader should make? Can the authors kindly add references to the ejection mechanism mentioned? For example, such scenario is advocated in Forbes&Romanowsky 2023 and Adamo et al 2023 to explain the position of the GCs in the Sparkler. But huge literature is available on the topic. 1 to 2 references would suffice.

Referees' comments:

1. Referee #1 (Remarks to the Author):

This paper presents new observations and an analysis of a lensed galaxy at high-redshift, detailing the physical properties of a possible Milky Way progenitor in the early Universe. The high-quality JWST imaging and spectroscopy alone provides tremendous value and would be worthy of publication on its own. I find the authors' analysis of the data and its implications for our understanding of galaxy formation to be thorough and robust. I have some minor comments and suggestions to make to improve the paper, which I detail below. Once those are addressed, I would recommend this paper for publication.

- 1.1. One general comment I have is that in several places throughout the paper, the magnification or magnification range is stated. Other quantities that depend on the magnification, such as the size, stellar mass, etc., are also presented in many places. These should either always include the magnification uncertainties due to uncertainty in the lens model, or it should be stated clearly the first

time the magnification is mentioned that the quantities either do or do not include them. It is not always clear whether some of the physical quantities presented in the paper factor in these uncertainties in the magnification.

We thank the referee for identifying this confusion. We have updated the text to reflect that the quantities always include uncertainties from magnification wherever applicable (sizes, stellar masses, etc.) unless otherwise stated (e.g. in the μM^* numbers in Table 2). We have also added “demagnified” in front of all stellar masses and size upper limits in the paper.

- 1.2. Figure 2, top right panel: Are Slit 1 and Slit 2 mislabeled in the 2D spectrum? The top trace (currently labeled "Slit 2") shows emission lines that appear in the 1D spectrum, even though the 1D spectrum is supposed to be Slit 1.

We thank the referee for spotting this oversight. The slits were indeed mislabelled in the 2D spectrum. We have fixed it in the new version of Figure 2.

- 1.3. Figure 2, bottom left panel: It's hard to evaluate this χ^2 map and what region of parameter space is allowed by the fit. Is it possible to make this a contour plot showing 68% and 95% quantile contours instead?

We thank the referee for this excellent suggestion. We increased the parameter grids in the nebular continuum fitting and updated the panel to plot the likelihood instead of the χ^2 values for visibility. We also overplot contours corresponding to 1-, 3-, 5-sigmas to show the parameter space that is allowed by the fitting. This has significantly improved the presentation of this figure.

- 1.4. Figure 2, bottom right panel: Is this plot for Slit 1 and Slit 2 combined?

The spectra on the bottom right panel is for Slit 1 only. The spectra for Slit 2, as well as for BF and NBF are shown in Figure 8 in the Methods section. We thank the referee for identifying this confusion and we have labeled "Slit 1" of the bottom right panel of Figure 2.

- 1.5. Figure 3, top right panel (and other similar panels throughout the paper): The x-axis is a bit unclear. Is time in the +x direction actually time prior to the source redshift? The axis label and/or caption should more clearly reflect this, or add a second x-axis showing redshift.

The x-axis in that plot is lookback time (i.e. $t=0$ is the time of observation at $z=8.31$ and $t=0.61$ Gyr is the age of the universe then). The axis and captions have been updated to reflect this, and we thank the referee for pointing this out.

- 1.6. Figure 3, bottom panels: The grey arrows seem unnecessary since the plots are clearly numbered. The arrow from 6 doesn't seem to be pointing to the correct object.

We agree with the referee that some of the arrows were mislabelled and that the figure will be cleaner without them. Hence, we have removed the arrows from Figure 3 (now Figure 4) and we thank the referee for the suggestion.

- 1.7. Figure 4, right panel: It seems like the x-axis range could be made smaller so we can see the distribution of the clusters better. Is there some easy way to indicate which clusters are which on this plot?

We have removed this figure from the paper altogether and have replaced it with a surface density vs. size plot. This was done to address the valid concerns raised by referee 3 about the assumptions made in the calculation of the density.

- 1.8. Figure 5: Indicate the size of the cutout in the caption or on the image itself. Does the circle in the bottom left of the images represent the PSF size? This should be stated in the caption.

The size of the cutouts are $10'' \times 10''$. The circle on the lower right corners are the FWHM of the PSFs. The captions have been updated to reflect this, and we thank the referee for pointing this out.

“The cutouts used for the fitting have size $10'' \times 10''$. In this figure we have zoomed in on the central $7'' \times 7''$. The FWHM of the point spread functions of the respective filters are shown as black circles on the lower right corner.”

- 1.9. Section 1.2, 2nd paragraph: When fitting the light components, are any of the parameters linked across the various filters, or are they all fit independently?

The center position of the sparkles are first fitted in F150W filter. The position, size, axis ratio, and the bending of the diffuse component is fitted in the F444W image. We then rerun the photometry with fixed centers of all components, and fixed size, axis ratio, and bending mode of the diffuse component. The free parameter for the point source and the arc is the flux only. We thank the referee for identifying this lack of description. We have updated the Methods section to reflect this information.

“For the morphological fit, we create $10'' \times 10''$ postage stamps in all ten filters from the BCG-subtracted images. We determine the priors for the centers of the ten clusters by visual inspection. While nine out of the clusters appear as point sources, FF-4 has an elongated shape and appears unresolved. We first determine the central coordinates of the clusters and the arc by fitting:

- i. an elliptical Gaussian for FF-4,*
- ii. nine point sources for the other nine clusters, and*
- iii. another elliptical Gaussian with the bending mode turned on for the diffuse arc to the F115W image, which has the highest resolution (smallest PSF).*

The free parameters are the centers and total fluxes of all the components, the radius and axis ratio of FF-4, and the radius, axis ratio, and bending mode (B2) of the arc. The initial guesses for the coordinates were determined by visual inspection of the F115W image. Once we obtain the fitted central coordinates of all the components from F115W, we again fit all eleven components in F444W, which has the highest signal-to-noise ratio for the arc and FF-4, to determine the radius, axis ratio, position angles of the ellipses, and the bending mode B2 of the arc.

We use the best-fit center coordinates from F115W as the central coordinates in all the filters. However, instead of fixing the central coordinates, we allow GALFIT to fit for them in every filter within a very narrow range of ± 0.5 pixels ($0.02''$) to account for the uncertainty in the PSF center. We also fix the bending mode B2 (2.14), ellipse radius ($3.9''$), axis ratio (0.08), and position angle (-51.8°) of the arc from the F444W fit. We fix the morphology of FF4 as well with (radius= $0.59''$, axis ratio= 0.1 , position angle= -53°).

We now fit all eleven components in all ten filters to determine their fluxes. The resulting models and residuals are shown in Fig.~1. Residuals from the fits are negligible, as shown by $\chi^2/\nu \sim 1$ in the GALFIT fits in all filters. This confirms the original visual impression that nine of the clusters are unresolved and an additional smooth component is present."

- 1.10. Section 1.2, 3rd paragraph: "...we inject the full Firefly Sparkle model in 100 random locations in our 10"x10" postage stamps..." - do you mean that you inject the model in random locations in the entire field, not the postage stamp? How large is the entire field?

We inject the Firefly model in 100 random positions in the 10"x10" postage stamp. We have updated the Methods section with this information.

"To derive the uncertainty in our flux estimation, we inject the full vffs model in 100 random locations in our 10"x10" postage stamps (avoiding the edge) and refit with the exact same setting of GALFIT."

- 1.11. Section 1.3.1: When deriving the total oxygen abundance of $12+\log(\text{O}/\text{H}) = 6.99+0.15/-0.33$, does this account for the different $T_{\text{e},\text{O}}$ measurements and their uncertainties? What about the assumption of n_{e} ?

We used the mean value of the two different temperature measurements in the oxygen abundance calculation, and the uncertainty is taken from the full range of the 1-sigma errors of the two measurements. We fixed the electron density n_{e} to 10^3 cm^{-3} , which is consistent with recent JWST observations of similarly high-redshift galaxies (e.g., Isobe+23). We have updated the method section to reflect this information.

- 1.12. Section 1.3.1: "Both ratios provide similar limit of $\log(U) > -2.1$ " - you haven't defined U yet.

Thank you for this oversight, we have now defined U when first mentioned.

- 1.13. Section 1.4, last paragraph: In determining the properties of each cluster, how much do the results change if you don't fix the mass slope or change the assumed value to something else? Why do you assume $\alpha = 1.0$ instead of the central value of $\alpha = 1.3$ from the fit to the data?

We thank the referee for bringing up this point, and have updated our fits in the paper to leave the high-mass slope of the IMF as a free parameter in all our fits. This leads to larger uncertainties in the mass estimates, but fully marginalizes over possible variations in the IMF slope.

We had originally used 1.0 instead of 1.3 to account for the maximum a-posteriori value instead of the median value, but show that leaving alpha free does not change our results significantly. This has also been updated in the methods section of the paper.

- 1.14. Table 2: Related to the previous point, it should be stated in the caption what assumptions are made here, and what the uncertainties actually reflect. The error bars seem quite small if all uncertainties (magnification, IMF, etc.) are taken into account.

The referee raises a valid point. As mentioned in the previous point, we have now significantly reduced our modeling assumptions, and marginalized over the full parameter space used for the spectrophotometric fits even while fitting the photometry, with larger uncertainties to show for it.

- 1.15. Table 2: Why isn't there a magnification for the FF-arc? While the magnification is different at various locations along the arc, you can presumably calculate a total flux to de-lensed flux to get the magnification. Is this how the other quantities for the FF-arc are derived? The same can apply to BF and NBF.

We measured the average magnification of the FF-arc by using the GALFIT model of the arc (in F200W) and selecting all pixels with flux >10% of maximum flux. We then computed the best model magnification value for all selected pixels and computed the mean and std value for these. This figure shows the magnification along the arc:

We have described our procedure in Method section 5:

“We measured the average magnification of the FF-arc by using the GALFIT model of the arc (in F200W) and selecting all pixels with flux >10% of maximum flux. We then computed the best magnification value for all selected pixels and computed the mean and standard deviation value for these to find the magnification of the arc ($\mu=24.4\pm 6.0$).”

- 1.16. Section 1.5, 1st paragraph: Were there any new multiply-imaged systems discovered in the JWST photometry? If so, why were they not used for the modeling? Have there been any independent mass models of this system?

No new multiple image systems were used to constrain the model compared to the literature available at submission time. However, all multiple images have been assigned a spectroscopic redshift derived from NIRSpec and NIRISS data. Since submission, an independent lensing model of the cluster constrained only with HST and MUSE data has been made public (Patel et al. 2024 arXiv:2405.04577) for which the magnification obtained at the Firefly position is compared. Lower magnification value was found with this model, however, this is due to the absence of a nearby cluster member in the lens list.

- 1.17. Section 1.7: In the main paper text, it is stated that the stellar mass of the Firefly Sparkle is $\sim 10^{6.8} M_{\text{sun}}$ and that it is consistent with the stellar masses expected for a MW or M31 progenitor. This section describes the procedure, but doesn't give any values for the range of stellar masses derived for such progenitor. This range should be given here for comparison, and to give a sense of the uncertainties in the procedure.

We thank the referee for pointing out this oversight and we have added the ranges for the stellar mass of both a MW and a M31 progenitor at the relevant redshift in paragraph 3 of Sec. 7. These ranges are $10^{5.71} - 10^{7.14} M_{\text{sun}}$ and $10^{6.13} - 10^{7.75} M_{\text{sun}}$ respectively. The Firefly Sparkle's total stellar mass of $10^{6.8} M_{\text{sun}}$ is within both of these stellar mass ranges. We have added this in Method section 7:

“At $(z=8.3)$, the median stellar mass of Milky Way progenitors is $(\log(M_{\text{star}}/M_{\text{dot}}) = 6.4 \pm 0.7)$ and the median stellar mass of M31 progenitors is $(\log(M_{\text{star}}/M_{\text{dot}}) = 6.9 \pm 0.8)$. The \textit{ffs} with a stellar mass of $(\log(M_{\text{star}}/M_{\text{dot}}) = 7.0^{+1.0}_{-0.3})$ is firmly within the 1σ stellar mass range of both Milky Way and M31 progenitors. More details on the progenitor matching technique can be found in Tan et al. 2024.”

2. Referee #2 (Remarks to the Author):

I have read the paper “The Firefly Sparkle: The Earliest Stages of the Assembly of a Milky Way-type Galaxy in a 600 Myr Old Universe” with interest. This paper presents a joint photometric and spectroscopic study of a gravitationally lensed arc at $z=8.3$, which exhibits a clumpy structure. The clumps sizes and stellar masses are consistent with early proto-globular clusters, and the spectra show nebular features that suggest a top-heavy initial mass function. The authors conclude that this galaxy is a likely progenitor for Milky-Way like galaxies. These results are compelling and warrant publication, however several issues will need to be addressed before I consider this manuscript to be ready for publication. I list major and minor points below.

Major:

- 2.1. The description of the NIRSpec data reduction is insufficient as currently presented. NIRSpec prism data is background limited, and the continuum shape is used in later analysis, so proper background subtraction is critical to ensure that the final result is not being influenced by leftover sky background. In its current state, the paper cites Asada et al. in prep. for the custom background subtraction method. With the Asada paper not available, it is not possible to assess the reliability of background subtraction, and thus not possible to assess the reliability of any of the continuum modeling described in the subsequent section. The authors either need to make the Asada paper public (an arxiv posting is sufficient) or they need to include the relevant details of the custom background subtraction in this paper.

We have clarified details of the NIRSpec reduction in the text, and thank the referee for bringing it up. The full details of the reduction are now in the manuscript text in Section 3, and we apologize for any confusion caused by the initial phrasing - while Asada et al. will present more detailed diagnostic plots, we have confirmed that the custom background subtraction method works as well as a standard drizzle background subtraction method used in e.g. Strait et al. 2023 and have elaborated on the exact procedure in the methods. The updated text now reads:

We used the standard JWST pipeline for the level 1 processing where we obtained the rate fits files from the raw data. We enabled the jump step option expand large events to mitigate contamination by snowball residuals, and also used a custom persistence correction that masked out pixels which approach saturation within the following 1200 s for any readout groups. We then used msaexp to do level 2 processing where we performed the standard wavelength calibration, flat-fielding, path-loss correction, and photometric calibration, and obtained the 2D spectrum before background subtraction. Since the central and upper shutters contain different clusters (see Fig. 2 top left to find the shutter positions), we need a custom background subtraction to avoid self subtraction. We did this by building the background 2D spectrum by stacking and smoothing the sky spectrum in the

empty pixels, and obtained the background subtracted 2D spectrum of Firefly Sparkle. We confirmed that this custom background subtraction method works as well as a standard drizzle background subtraction method used in literature (e.g., Strait et al, 2023), using a well-isolated galaxy spectrum from the CANUCS observation (Asada et al. in prep.). We finally extract the 1D spectrum separately in Slit 1 and Slit 2, by collapsing the 2D spectrum using an inverse-variance weighted kernel following the prescription by Horne (1986).

- 2.2. I am somewhat concerned about the [OIII]1660,6 and [OIII]4363 detections, and thus the validity of the unusually high electron temperature measurements. Based on the line flux uncertainties quoted in Table 1, each of these are detected at ~ 4 sigma. However, looking at the spectrum in Figure 2, I see several features that are not fit as emission/absorption lines that appear (by eye) to have similar SNR. Could the authors please define the criteria by which emission features were selected.

We thank the referee for bringing up this point, and have clarified it further in the text (lines 482-514). The S/N values in Table 1 represent uncertainties on integrated emission line fluxes from Gaussian profile fitting. The [OIII]4363 line peak is 9.5 sigma above the continuum, significantly higher than other unfit features. Its relatively large flux uncertainty stems from deblending it from H-gamma, not raw spectral S/N. Using a "5-sigma above continuum" criterion would not select any additional emission line candidates. While the [OIII]1660,6 detection is tentative, it serves as an independent consistency check for the [OIII]4363 measurement. The temperature determination doesn't solely rely on this line. Additionally, we have a third independent measurement of the high electron temperature from modeling the nebular continuum spectrum, and find all three estimates to be consistent within uncertainties.

re: unidentified features: we identify emission lines by examining both 1D and 2D spectra for typical features at expected wavelengths, eliminating spurious detections through visual inspection of the 2D spectrum map. Real features span multiple pixels in both directions due to PSF and LSF effects. For example, while we see a peak at 4.3 um in Slit 1's 1D spectrum, that small peak is due to a single noisy pixel in the 2D spectrum resulting in a 1 pixel wide peak in the 1D spectrum, and is thus not a real feature.

- 2.3. Related, it has been found that the NIRSpec reduction pipeline underestimates the uncertainty of the final spectra (see e.g. Appendix B, <https://ui.adsabs.harvard.edu/abs/2023ApJ...951L..22A/abstract>). It is not clear how the uncertainty spectra are calculated with msaexp. Could the authors please clarify how the uncertainty spectra are calculated, and whether or not they are similarly underestimated?

The uncertainty spectra are derived from the 2D inverse variance map, which msaexp generated from the level 1 products by the custom STScI JWST pipeline. To investigate whether the uncertainty spectra are properly estimated we did the following test:

1. Extract the 1D sky spectrum and its associated error in the empty shutter (i.e., the bottom shutter of the FF slitlet) in the same way as we do for science spectra;
2. Normalize the sky spectrum by the uncertainty array to obtain the noise-normalized sky spectrum;
3. Compute the fraction of sky pixels having <1 -sigma and <2 -sigma levels in the noise-normalized spectrum.

If the uncertainty spectrum is properly estimated, the noise-normalized sky spectrum should follow a normal distribution. We found that 66% and 94% of sky pixels have <1 -sigma and <2 -sigma, respectively, which is fully consistent with a normal distribution. It is possible that the uncertainty is slightly underestimated because 1-sigma and 2-sigma should contain 68.3% and 95.5% of the total

ideally, though, only a 4% increase of the noise level is required to reconcile this. We thus conclude that the uncertainty spectra are not significantly underestimated.

We have added this in the Method section 2:

“We verified that the uncertainty array of the 1D spectrum has the appropriate normalization by testing the distribution of spectral fluctuations in an empty sky region and finding the fractions of pixels at $>1\sigma$ and $>2\sigma$ was as expected.”

- 2.4. The text states that FF-4 “exhibits an elongated component”, and is thus fit with a Gaussian ellipse. However, the major axis is later stated to be smaller than the PSF, indicating that the object is unresolved. These two statements appear to be at odds with each other - an unresolved clump should look like the PSF, and thus should not appear elongated. Looking at Figure 5, it appears to me that the clump is indeed unresolved. Could the authors please either (1) edit the text to consistently refer to this clump as unresolved, or (2) provide additional details (e.g. from the GALFIT modeling) to show that the clump is partially resolved?

FF-4 indeed exhibits an elongated component and had been fitted with a Gaussian ellipse. We thank the referee for noting the discrepancy in text. We have updated the text to provide the morphology parameters of FF-4:

“We determine the priors for the centers of the ten clusters by visual inspection. While nine out of the clusters appear as point sources, FF-4 has an elongated shape and appears unresolved. We first determine the central coordinates of the clusters and the arc by fitting:

i. an elliptical Gaussian for FF-4,

ii. nine point sources for the other nine clusters, and

iii. another elliptical Gaussian with the bending mode turned on for the diffuse arc to the F115W image, which has the highest resolution (smallest PSF).

The free parameters are the centers and total fluxes of all the components, the radius and axis ratio of FF-4, and the radius, axis ratio, and bending mode (B2) of the arc. ...

We fix the morphology of FF4 as well with (radius=0.59", axis ratio=0.1, position angle=-53°).”

- 2.5. The authors state in Methods Section 1.2, “The agreement between NIRISS and NIRCcam fluxes in the three overlapping filters is another confirmation of the robustness of photometry”. However, looking at Table 3 and the bottom panel of Figure 9, it seems like many of the overlapping filters are in significant tension with each other (some at the 3 sigma level or greater). This would seem to contradict the statement in the text. Could the authors please explain this tension?

We acknowledge the referees concern about the discrepancy between NIRISS and NIRCcam fluxes in the three overlapping filters. This was an issue stemming from the reduction of NIRCcam images using grizli. The NIRCcam images had sigma clipping of bright stars which resulted in erroneous characterization of PSF. Since we were fitting our point sources with the empirical PSF, the issue trickled into our analysis. We are happy to report that this issue has since been resolved and the NIRISS and NIRCcam fluxes in all CANUCS fields are in agreement with each other in our new image reduction (v2p0). We redid our photometry using the exact same method as described previously on the new images using the new empirical PSFs and we find our overlapping fluxes are in better agreement, within the uncertainty. Below is the new Figure showing the agreement in fluxes.

- 2.6. Could the authors please elaborate on their reasoning for fixing certain parameters in the photometric SED fits for all star clusters based on the spectrophotometric results that cover only a subset of the clusters? It is stated that clusters 3-6 are covered with the spectroscopic observations. Based on the source plane model shown in Fig. 3, cluster 7 appears close enough to be associated with clusters 3-6, but clusters 1,2, and 8-10 seem to be far enough away that it is not clear they will have consistent properties.

In response to this comment and suggestions from the other referees, we have refit the photometry for the individual clusters varying the same set of parameters as the spectrophotometric fits, marginalizing over all the parameters that these fits are not able to constrain. This leads to larger uncertainties, but accounts for possible variations in these parameters.

The motivations for fixing the IMF, ionization and dust and adopting a physically motivated prior on sSFR while fitting the clusters are twofold: (i) these quantities were generally not expected to vary significantly across the galaxy, and (ii) the SNR and information content of the photometry is significantly lower than the NIRSPEC spectra used to estimate the IMF and ionization parameter. Indeed, varying these quantities returns a flat posterior consistent with sampling the prior for most of the clusters, so we have fixed it to default values. For dust, we have run fits with and without varying the dust prior and found that the fits are generally consistent with low dust. Hence given the balmer decrement it is likely that the galaxy is largely dust-free, hence the rationale behind fixing dust. Lastly, the sSFR priors, while motivated by the fits to the NIRSPEC data, are wide and still allow for a diverse range of star formation histories, as evidenced by the histories of clusters 8 & 9, which show stellar populations and sSFR quite different from the others. We have elaborated on our fitting considerations in section 4 of the text and thank the referee for bringing this up.

Minor:

- 2.7. Main text, paragraph 2 - please specify how the range of magnifications is determined (based on table 2, it seems this is a range of best fit magnifications, but does not account for uncertainties?)

We thank the referee for noting this oversight in text. We have updated the Methods section with the following paragraph:

“Uncertainties on the magnifications are computed from 100 randomly selected models from the optimization after its convergence around the minimum χ^2 . The numbers provided in Table 2 are the median and $\pm 1\sigma$ limits on the distribution of the 100 values obtained at the position of each cluster.”

- 2.8. Fig 2 - To my eye, the emission lines in Slit 2 in the 2D spectrum appear brighter than those in Slit 1, however your extracted 1D spectra show the opposite. Could you more clearly delineate the 2D spectra of Slit 1 and Slit 2? I trust your 1D extractions, I'm just struggling to interpret the figure.

We thank the referee for spotting this oversight. The slits were indeed mislabelled in the 2D spectrum. We have fixed it in the new version of Figure 2.

- 2.9. Page 7, first full paragraph - "...only spectrum with SNR high enough to constrain the slope" - I'm guessing "the slope" here refers to the IMF slope? Please clarify this in the text

This sentence has been rewritten due to other changes in the manuscript, but yes, this refers to the IMF slope and has been clarified throughout the text.

- 2.10. Page 8 - "given the extreme shear...it is unlikely that the magnification will significantly change" - This seems backwards to me. Typically objects with high magnifications have high uncertainties, yet this sentence appears to indicate the opposite. Do you mean that the large shear implies the magnification will remain high as the lens model is refined?

We thank the referee for identifying this mistake. We have removed this sentence from the manuscript.

- 2.11. Methods Sec. 1.2, Paragraph 2 - Could the authors please elaborate on their construction of an empirical PSF? Which stars are used/what criteria are used to select them?

We thank the referee for noting this oversight in text. We have updated the Methods section with the following paragraph:

"Point spread functions are extracted empirically by median stacking bright, isolated, non-saturated stars following the methodology described in Skelton et al. (2014). Convolution kernels for homogenizing all data to the F444W resolution are created with photutils.psf.matching using a SplitCosineBellWindow() windowing function to remove high-frequency noise, which results from floating-point imprecision when taking the ratio of Fourier transforms. We optimize the shape of each window function to minimize the median residual between convolved stars from each source filter being convolved and stars from the target F444W filter."

- 2.12. Methods Sec. 1.2, end of Paragraph 2 - "Residuals from the fit are negligible" - By what metric are the residuals determined to be negligible?

We use the χ^2/ν of the GALFIT fits to determine the goodness of fit. For all individual filter fits, the χ^2/ν value range between 0.95 to 1.05, i.e. very close to 1. We have updated the text to include this information:

"Residuals from the fits are negligible, as shown by $\chi^2/\nu \sim 1$ in the GALFIT fits in all filters."

- 2.13. Methods Sec. 1.2, Paragraph 3 - "we find no significant systematic offset..." Please quantify the offset, and state which metric was used to determine its insignificance.

The offsets are calculated as follows: (injected flux - recovered flux)/(error in recovered flux). The "recovered flux" is the mean (biweight location) of the 100 measured fluxes. The error is the standard deviation (biweight scale) of the 100 measured fluxes. As can be seen from the figure

below, there is no significant difference between any of the injected and recovered fluxes for all 11 components in all filters.

2.14. Section 1.3 - Were all objects (FF, BF, NBF) observed in all 3 MSA configurations? And thus was each object observed with the same exposure time?

Yes, each of the targets were observed in 1 MSA configuration, with the same exposure time of 2889 seconds.

2.15. Fig. 7 - The caption here does not provide much information about the figure. Please expand this figure caption.

We have updated the caption of Figure 7 to provide more clarity. The new caption reads as follows:

Figure 7: Inferring physical properties from the spectrophotometric fits. The top panel shows the spectrophotometric data for slit 1 (black line and points) along with the best-fit spectrum from Dense Basis (orange line and points). An inset panel shows the fit in the region of H β + OIII, where the spectrum has much higher fluxes). The corner plot in the bottom shows the posteriors for each parameter being fit, showing fits to only photometry (light blue contours), only spectroscopy (red contours) and both (black contours). The contours show the 1- σ and 2- σ regions for each posterior, along with their covariances, while the diagonal plots show the marginal posteriors for each parameter. In addition to the joint posteriors, the spectra and photometry posteriors generally agree, with the spectra better able to constrain parameters like the gas-phase metallicity. The inset panel in the middle left shows the stellar population posteriors from photometry alone (light blue), spectroscopy alone (red) and joint (black lines) along with 1- σ uncertainties, again finding good agreement between photometric and spectroscopic fits within uncertainties.

2.16. Page 15 - please cite the solar abundance determination used.

We have revisited several references and found that the solar abundance of $12+\log(\text{O}/\text{H}) = 8.69$ is more common than 8.6. We switched to use 8.69 instead, and provided a reference of it.

2.17. Sec 1.4, first paragraph - the prior on IMF high-mass slope appears incomplete?

We thank the referee - the updated text now reads: 'with a flat prior for the high-mass slope α in $[1., 4.]$ '.

- 2.18. Sec. 1.4, last paragraph - Please specify which IMF high-mass slope value is used for the photometric fitting. The current text ($\alpha \sim 1$) is not clear.

Similar to responses 1.13 and 2.6, we refit our photometry to include the whole range of possible IMF values used for spectrophotometric fitting, with larger uncertainties that reflect the effect of marginalizing over this parameter.

- 2.19. Sec. 1.5 - If the lens modeling paper (Desprez et al.) is not publicly available, please include a figure showing lens model constraints and the lensing critical curve.

Here we include a figure displaying the cluster central part with the position of all the multiple images used to constrain the model and the resulting magnitude 50 iso-contours for sources at redshift $z=8.3$.

3. Referee #3 (Remarks to the Author):

Dear authors,

I have read with great interest the manuscript "The firefly Sparkler: The earliest stages of the Assembly of a Milky Way-type galaxy in a 600 Myr old Universe". The manuscript presents the spectra-photometric analysis of a remarkable lensed galaxy at spectroscopic redshift of 8.3. The Firefly sparkler (FF) is an interacting system with at least 2 more galaxies located at similar redshift. I have several major concerns

that affect many of the key conclusions the authors reach. In many points, the manuscript lacks clarity or connection between the different topics. Important aspects connected to the conclusions drawn in the manuscript are in the methods, while text that is not relevant is included in the main text. Many statements are made and not refereed. Many assumptions made in the analysis but are incorrect and affect the entire analysis and interpretation of the results.

- 3.1. It is difficult for the reader to read through the main story the article is aiming at. The summary and main text of the article jumps from one topic (star clusters) to the other (ionisation status of the ISM of the galaxy), with a third topic (potential interaction with other galaxies) making the appearance, but there is no clear connection between the three topics in the text. In my humble opinion, the connection throughout the text should be the star clusters. The authors have the unique, truly unique, opportunity to link cluster feedback at physical scales of a few parsecs to the extreme ISM conditions that JWST is revealing in many of the reionization era galaxies. I believe that that is the real breakthrough of this object and work. However, the authors dwell into globular cluster formation and Milky Way progenitors, which I actually find weak and circumstantial at best (a lot of speculation and little results). The authors misuse SED fitting techniques to fit star clusters. A galaxy can have a non-parametric SFH, a star cluster in local or high redshift universe is a stellar system of a few parsec scales that form in an instantaneous event. We have 30 years of literature to support that, both from numerical and observational constraints, and this cannot simply be ignored. The authors at some point mention in the text that only non-parametric SFHs can reproduce elevated EWs. Well, this might be true for galaxies, but actually simple stellar populations produce normally (even in the local universe) EW of 1000 Å. Therefore, I urge the authors to use the information at hand, i.e., detecting bound star clusters to actually use SFH that assume a delta burst. This can be done with a short exponential tau of 1 or 10 Myr, a burst assumption, a SSP. Moreover, it is very important to not assume an underlying continuous star formation history. The way photometry has been performed ensures that the diffuse galaxy light has been removed from the cluster light. Once the authors make this change in the assumed SFH to fit the cluster light they will find extremely young ages and finally quantify the amount of feedback that each of these little sparks are producing. I expect that the cluster producing the ionisation detected in slit1 is about 3 Myr old. The other clusters will show age gradient moving away from this central cluster. I would encourage the authors to include citations to work done at lower redshift where cluster feedback has been evaluated. They should include citations to Toppings et al 2024, Vanzella et al 2023 (Sunrise arc), Vanzella et al 2023 (Lap1), any of the many works done in the sunburst arc, the first confirmed star cluster leaker at redshift 2.4.

The referee raises a valid point, and we have significantly reworked the main text of the manuscript to refocus our results and performed extra analysis to explore the scenarios suggested by them. The original motivations behind using SED fits with non-parametric SFHs were that (i) the fitting was flexible enough to account for both sharp recent bursts that are effectively SSP-like and extended episodes of star formation (ii) an accounting of possible residual light from the diffuse component that bleeds into the photometry due to PSF and other effects. Many simulations have also suggested that star clusters do not universally have SSP (for example, Krumholz and Tan 2007, Matzner 2017, Rieder et al. 2022, Karam and Sills 2022)

In response to the referee's suggestion, we have now (i) redone our photometry as described in section 2 of the paper, and (ii) redone our fits with SSPs as well, and now report values for masses and ages of the individual star clusters. While we see a mild indication of an age gradient with the new ages (see figure below), we feel that it is not strong enough to include in the paper, and will be pursued with higher SNR data in future observations:

Although the t_{age} from SSP and t_{50} from DB fits (see Table 2) may appear inconsistent, it is important to note that the DB fits for most star clusters indicate a sharp burst of star formation within the last 10 million years (see Figure 10). By design, an SSP is biased towards this recent burst, whereas a non-parametric SFH can accommodate extended episodes of star formation. However, with our current data, we cannot distinguish between extended SFH in the star clusters and the contribution of light from the diffused arc.

We have also included citations to relevant work and thank the referee for their suggestions.

- 3.2. Regarding all the circumstantial results collected to probe the IMF shape. None of them reads convincingly. I bring up the work done by Cameron et al 2024, which has been widely referred to in the manuscript. Cameron and collaborators find indeed galaxies that show a Balmer jump and a steep turnover at $<1430 \text{ \AA}$. Both features point toward an elevated electron temperature as well as the need of a very hot stellar spectrum, which combined with the nebular continuum can explain the strengths of the turnover in the FUV.

Differently to what done by Cameron and collaborator, the authors do not include any treatment of IGM absorption (or a DLA in front of the galaxy). There is no mention of it throughout the analysis description. I don't need to explain that the light of a galaxy at redshift 8.4 will still be affected by neutral hydrogen absorption, either in proximity of the targets or along the line of sight. That such

absorption is playing a role could be seen in the absence of Ly α emission (even if the latter could be missing because of the prism resolution, although notice the booming optical emission lines...). It is very important that the authors produce clear evidence of a turnover due to extreme ISM and ionisation sources after IGM and DLA absorptions have been taken into account.

The second necessary step before claiming top-heavy IMFs is to use correct stellar population models in the SEDs. Again, the authors appear to ignore that in star clusters the most massive stars that can be produced reach easily 300 Msun, moreover stripped stars could provide a hot ionising source. The authors have so far assumed a typical Kroupa IMF. They do not even specify what upper-mass limit cutoff they use in the fit. Hence, I would like to see fits where the IGM absorption is accounted for and normal IMF with higher mass cutoff and binary populations (like those produced in BPASS) are used. It remains unclear and it is not explained in the text why the authors fit all the spectrum, but then when they fit photometry and spectrum combined, they stop well before the IGM absorption part (and exclude the F115W). This is not explained in the text although it is done in the analysis presented in Fig 7 and 9. The observed SED is not done on 10 bands but effectively on 6 since the NIRISS data do not provide any further constraint than NIRCcam filter and the bluest filter is removed.

We thank the referee for their thorough and insightful comments, and agree that additional analysis is necessary to strengthen our conclusions. We have revised our analysis to include IGM and potential DLA absorption (described further below), explore more comprehensive stellar population models including those with binary populations, and present further details on our spectral fitting methodology. We considered dropping the section about the IMF altogether, but retained it in light of (i) the combination of the Balmer jump and the emission lines being able to provide constraints on the top-heavy IMF slope, (ii) the signal persisting between the DB and SSP fits, and (iii) the relatively poorer fits with the BPASS models. We stress that the spectrophotometric fits in the paper only consider the portion of the spectrum blueward of the turnover, and thus are relatively unaffected by modeling choices of the IGM absorption and DLA, but have included them for completeness.

In terms of the turnover, the IGM/DLA absorption should be negligible given the blue continuum and the sharp drop out in the slit 2 spectrum. Attached is the zoom-in spectra around the Lyman break comparing Firefly slit 1 (red) and slit 2 (blue). We also show the 1D spectrum of the bluest trace within the slit 2 (black) in this plot. All spectra are normalized at 2.2 μ m for clarity. The slit 2 spectra (blue and black) are rather blue and have sharp Lyman breaks starting at 1.2 μ m (dotted line), while the slit 1 spectrum (red) has the turnover starting at \sim 1.4 μ m. Assuming that the slit1 and slit 2 spectra have the same IGM/DLA absorption (which is plausible considering their physical proximity), the IGM/DLA absorption can affect only at 1.2 μ m and bluer, and the turnover feature at \sim 1.4 μ m in slit 1 spectrum cannot be due to the IGM/DLA absorption.

Nevertheless, to avoid possible effects by IGM/DLA contributions, we have redone the nebular continuum fitting with masking out at $\lambda < 1.2 \mu\text{m}$. Figures, numbers and text are updated accordingly but have not changed significantly. We also want to note that the spectrophotometric fitting in Section 4 only used $\lambda > 1.5 \mu\text{m}$, and is not affected by the shape of the Lyman break.

We have also updated the text to include fits using the BPASS models implemented in FSPS (“bin-imf135all 100”, assuming a Salpeter IMF with an upper mass cutoff of $100M_{\odot}$), performing fits with both SSP and DB star formation histories using these models. While Table 2 details the physical properties derived using these models, they generally show worse χ^2 values and are not able to as accurately model the spectra using the canonical IMF implemented. The varying Kroupa IMF used with MILES+MIST fits has an upper mass cutoff of $120M_{\odot}$, and although varying this to $300M_{\odot}$ changes the overall normalization making the clusters heavier, it does not significantly affect the overall ages. The IMF posteriors with three different scenarios using MILES+MIST are shown below, which all agree within uncertainties and indicate a top-heavy high-mass slope. We also show ages from the three fits using SSPs for reference, finding no strong trend from modeling choices.

3.2.1. Since assumptions from the spectrophotometric analysis are propagated to the fit of the clusters, the latter analysis does not provide any constraint on the IMF shape.

The referee raises a valid point, and we have addressed it by refitting the photometry without assuming any parameter values from the spectrophotometric fits. As before, the individual

clusters do not provide any information about the IMF shape since photometry alone can not constrain this (individual posteriors show a slight tendency toward top-heaviness but are not statistically significant).

- 3.3. Abundance matching method to link the FF to Milky Way assembly. This method is based in many assumptions and extrapolations. The authors do not have dark matter halo mass at hand and prescription to associate mass at this redshift are truly unreliable. Looking at the broad work presented in the literature that has tried to simulate star cluster formation and evolution during galaxy assembly (see, Grudic et al 2023, and all references therein for the very large literature!!) there is agreement to the fact that the Milky Way globular clusters with masses above $10^5 M_{\text{sun}}$ start to form at significantly lower redshift ($z < 7$) than the FF galaxy. If we rely on works that actually look at Milky Way-like disks, there is no evidence that the FF galaxy will be a progenitor of the MW. More likely it will be a progenitor of an early galaxy. In the text the authors write “We find that the Firefly Sparkle has a total demagnified stellar mass of $\log M \sim 6.8 M_{\text{sun}}$ ”. Does this estimate include the mass in star clusters? It is misleading since in the text the authors quote only the diffuse light mass, but the clusters are an important component in such young galaxy and cannot be neglected. Table 2 report some masses but not the magnification value used to derive the lensing-corrected mass. Please provide in the text the total mass of the galaxy and in Table 2 an entry for diffuse component and for total galaxy light (cluster + diffuse).

We thank the referee for raising this concern, indeed finding galaxies that match the Milky way's evolutionary trajectory can be an involved task, we have rephrased the text to refer to a typical galaxy (since the mass of this galaxy falls near the L^* value at this redshift) or a milky-way mass galaxy at most. The main point we want to make (as seen in the updated figure 3 in the main text) is that this galaxy is less massive compared to most of the galaxies observed at high redshifts, and is the only one resolved down to star clusters, which provides a unique opportunity to study the formation of galaxies more typical of what we see at lower redshifts. We have tried to illustrate this by adding previous JWST observations to this figure.

The total demagnified mass reported in the paper includes the masses of the individual star clusters, and has been clarified in the text and Table 2. We thank the referee for pointing out this confusion.

3.4. I will now follow the organisation of the text:

3.4.1. Summary. There are several statements which are not supported by the literature. For example:

“Composed today primarily of old stars and correlating with the properties of their parent dark matter halos, the first globular clusters are thought to have formed during the earliest stages of galaxy assembly.” Actually, this simplified idea is not that obvious and actually does not reflect what is known in the GC community. An idea of where we stand with understanding of GC formation in the Milky Way progenitor can be grabbed from the following work and reference therein. The lack of GCs with metallicities below the -2.5 floor suggests that the GCs in the Milky Way formed at $z < 6$.

<https://ui.adsabs.harvard.edu/abs/2019MNRAS.486L..20K/abstract>

The referee raises a valid point, and we have removed the relevant sentence from our discussion. It is also important to note that the metallicity floor of -2.5 in our fits comes from limits in our isochrone models rather than actual lower limits on the metallicity, which might be lower based on our estimates from the emission line ratios.

3.4.2. Summary. “The mass distribution of the galaxy seems to be concentrated in ten distinct clusters (~ 49 – 57% of the total mass), with individual cluster masses ($M_{*,\text{cluster}} \sim 105.3 - 105.8 M_{\odot}$) that straddle the boundary between low-mass galaxies and high-mass globular clusters.”. I’m not sure why is relevant to quote in the summary the ultra-faint dwarfs? Ultra-faint dwarfs have dark matter the star clusters that the authors are studying do not. I would rather find more significant/relevant a reference to the peak of the GC max function which is $\log(M) \sim 5.2 M_{\text{sun}}$ which is close to be constant in all GC populations studied at redshift 0.

The referee raises a valid point, and we agree that this point is not a fair comparison. We have removed this sentence from our discussion.

3.4.3. Summary. “The cluster ages suggest that they are gravitationally bound with star formation histories showing a recent starburst possibly triggered by the interaction with a companion galaxy at the same redshift at a projected distance of ~2 kpc away from the Firefly Sparkle.” How can single stellar populations show star formation history in their interiors that can tell a recent interaction has taken place in the galaxy? The age of the star clusters should be referred to. Unfortunately, with the current assumptions made in the analysis cluster ages cannot be used to trace the recent burst history of the galaxy and the propagation of star formation within the galaxy.

This has been addressed with our new fits, which use non-parametric SFHs to describe the galaxy as a whole and simple stellar populations (SSP) fits for the individual clusters, and only refer to the recent uptick in SFR on a galaxy wide scale for both the Firefly and its two companions.

3.4.4. “Due to its high [O iii] equivalent width (EW) contributing to the F444W broadband flux, this source can appear to be a double-break source similar to those in Labbe et al. 6 and Desprez et al. 7”. I believe that this is redundant to the context of the manuscript. Please remove.

The referee raises a valid point, and have removed this sentence.

- 3.4.5. GALFIT analysis. From text in main manuscript and text in the Method section is not clear how the photometry is derived. Do the authors allow the shapes of clusters and diffuse light to change from filter to filter or is fixed to the ones extracted in the F444W or F150W? I'm worried that if the authors let the shape to be fit in each band and allow all parameters free, they all be dominated by differences in the faint low S/N regions and therefore they include in each band different physical areas in the different components (clusters, galaxy light). This will result in differences in the recovered SED shape, thus affecting the reliability of all derived parameters. This is mostly concerning since by looking at the SEDs in figure 9. **The flux in the same filter but different instruments produce significantly different fluxes. Yet there is no mention in the text about this problem or clear explanation about the way the photometry is extracted.** Finally, I would like to stress that looking at the residuals in figure 5 significant light remains at the location of the clusters and visible in all the band separately and showing up in the color composite at the bottom-right.

We thank the referee for bringing up these fair points. We have responded to all of these issues in responses 1.9, 1.10, 2.4, 2.5, 2.7, 2.12, and 2.13. Our tests have shown that our method is effective at recovering the flux from point sources. We also have significantly less residue in f444w filter with the new version of our images and better characterized psfs, as can be seen below.

- 3.4.6. I find this statement redundant “Resolved photometry is also necessary for estimating the total mass of the Firefly Sparkle, as global spectral energy distribution (SED) fitting can bias stellar masses when a young stellar population outshines the first episodes of star formation (e.g., 9–11)”.

We have removed this sentence.

- 3.4.7. It seems from Table 32 and comments in the text that the fit to the diffuse light and that of the star clusters provide exactly the same output. On this point. It is not clear in Table 2 what values are reported. The star clusters have values that are corrected for magnification. While the galaxies do not have any magnification value quoted and therefore it is not clear if the mass reported for the 3 galaxies is corrected for magnification. Moreover, please notice that the total mass of the galaxy is the diffuse light+star clusters so I hope (but it is not clear from the text and tables) that that is what the authors quote for the galaxy mass. I think that it would be beneficial that the authors fit the total light of the galaxy+clusters to check whether they will recover similar results for the galaxy mass and SFR, SSFR.

We include demagnified masses of all ten star clusters and the FF-arc in our total stellar mass calculation. Table 2 provides the individual demagnified masses of the clusters from the 4 SED fits, Table 3 provides the magnifications and upper limits on size. We have also performed SED fits of the integrated photometry and find consistent results, as shown by the SFHs on Figure 4 and 10, as well the physical properties in Table 2.

- 3.4.8. Figure 2. In the top right panel, the authors write slit1 and slit2 but I believe it is inverted as from the text slit1 has the strongest lines that in the figure are labelled slit2.

We thank the referee for spotting this oversight. The slits were indeed mislabelled in the 2D spectrum. We have fixed it in the new version of Figure 2.

- 3.4.9. “These features are quite similar to those observed in [19–23], suggesting that the nebular continuum is a significant contributor to the overall SED for this star-forming cluster.” The cluster cannot be star forming if its age is larger than 100 Myr (e.g., please fit with the correct SFH assumptions). Moreover, as the authors notice later on, the light of multiple clusters enters this region, therefore I would refrain from calling the source within slit 1 cluster. I would encourage the authors to use the term stellar clump, ensemble of stellar clusters and stellar light in a region that is compact but not exactly coinciding with a single cluster.

We have reworded the sentence according to the referee’s suggestion. The new sentence reads: ‘suggest significant nebular continuum contribution to the overall SED for \ch{clusters contributing to slit 1}’

- 3.4.10. I cannot avoid to point out that I do not understand why the authors add 1 reference to Adamo et al 2024 work and only when discussing stellar densities. I believe that the comparison with that work is fundamental to understand similarities and differences. This manuscript presents the second highest evidence of star cluster formation in reionization era. The text of the manuscript touches on many points which are similarly discussed in Adamo et al., yet no reference is made to that work. Moreover, the concept of bound star clusters at high redshift has been proposed in the literature by Vanzella et al 2019-2020, then discussed in Metric et al 2022, Claeysens et al 2023 (the latter work also discusses the sparkler). No mention of any of the many works at high redshift that look for bound star clusters can be found in the text.

We have included the other references suggested by the referee, and thank them! We could not find the Metric et al. 2022 paper, however. In response to the referee’s other point, we have focused our discussion on the spectrophotometric properties of the Firefly sparkle, which we find difficult to compare with Adamo et al. 2024 since that object does not have a reliable spectroscopic redshift or measurements, which make it difficult to compare properties on a apples-to-apples basis. Since the star clusters themselves are studied photometrically in both papers, we have referred to that work while comparing their properties (ie. the stellar densities of the clusters in new Figure 3b (right panel)).

- 3.4.11. “Varying the power-law slope of the Kroupa IMF in FSPS results in an excess of high mass stars, which (i) increase the nebular continuum from the two-photon component [and therefore the strength of the Balmer jump; 19], and (ii) increases the equivalent width of the emission lines.” Single stellar populations can easily explain $\log(EW) \sim 1000$, no need to advocate top-heavy IMFs.

The referee raises a valid point. However we find that even with the SSP fits the method seems to prefer a top-heavy IMF, and hence have left this statement in.

- 3.4.12. “The motivation for the Firefly Sparkle came from the Sparkler [29] - a strongly lensed galaxy at $z = 1.38$ surrounded by old star clusters that could be resolved only with JWST. In contrast to the Sparkler, the Firefly Sparkle represents one of JWST’s first spectrophotometric observations of an extremely lensed galaxy assembling at high redshifts, with clusters that are in the process of formation instead of seen at later epochs.”. Honestly, looking at the currently very old ages of the star clusters in the FF galaxy there is not really support to the claim of being clusters in the process of forming.

We have removed this sentence.

- 3.4.13. Section 1.6 of Methods. I have to honestly say that I do not understand what is written in this section:

- 3.4.13.1. The authors use the main text to introduce all quantities that will be used to produce Figure 4 of the main text. Yet all the discussion of that figure and implications are done in Section 1.6 that reads like a few paragraphs extracted from the main text of a previous version with more figures and different analyses presented here.

We have removed the right panel of the figure 4 (now figure 3) and hence have removed the discussion on the density calculation.

4.

- 4.1.1.1. Why the authors show in Figure 3 and mention here a mass map? How is this mass map derived? Did the authors perform a px-to-px fit? There is no mention in the text. Why is this mass map relevant for the analysis? The authors mention the fit to the diffuse light of the galaxy and I understood from the text that is done extracting integrated fluxes.

We have added a few sentences on how the source plane model and the mass map was created. We have also shown both the RGB source plane model and the source plane mass map for easy comparison. We thank the referee for pointing out that not enough information was included on the mass map in the Method section.

“The source plane reconstruction is made using the best GALFIT model to compute the source plane positions and magnification for the ten star clusters. We use `\texttt{Lenstool}` to generate a source plane image reconstruction of the diffuse light of the galaxy with a smooth PSF-deconvolved model of its light profile. We use GALFIT to add ten point sources convolved with the appropriate PSFs to the diffused source plane model at the source plane positions of the star clusters with the demagnified fluxes. This process is repeated to generate source plane models in all filters. We also generate a mass map using the same prescription, replacing the demagnified fluxes with the demagnified masses. The resulting source plane RGB image and mass map are shown in the bottom panel of Figure 4.”

4.1.1.2. “We approximate an upper limit on the 3D half-mass radius from our projected upper limit on half-light size ($r_h = 4/3r_{50}$). The density is calculated using $\rho \equiv 3M/(8\pi r_h^3)$, where M is the total demagnified mass of the star clusters and the diffuse arc (assuming $\mu = 25$).” Why is the diffuse arc mentioned here?? “The resulting density vs. stellar mass plot is shown in left panel of Figure 4.” “No such plot is presented in the paper. “Here, the half-mass radius of the galaxy R is calculated from the mass map (shown in Figure 3.” —> how is the radius of the galaxy relevant to the estimate of the cluster density? Again, this is the mass map showed in Figure 3 and never introduced in the main text. The content of Figure 4 right is described as “we solve for Equation B6 in Gieles et al. [17], assuming $\zeta = 2$ for the early universe, and scaling the Milky Way rotation velocity to the mass of our galaxy.” How can the authors justify this for an early galaxy at redshift 8?? There is no way to possibly justify that this galaxy is a scaled version of the Milky Way rotation. Dwarf galaxies have very little rotation and almost no shear. The model presented by Gieles et al is exclusively developed for spiral galaxies and cannot be simply scaled down to dwarf galaxies at redshift 8. The dynamics and kinematics of these early galaxies are unknown. Even in the local universe, the Gieles et al relation would not be applicable to dwarf galaxies. “This figure is showing us that most of these star clusters are expected to survive to the present-day universe, and will expand and then get ripped apart to form the stellar disk and the halo of the galaxy. The only way they survive is to get kicked out to large distances, away from the dense tidal field of the galaxy.” while in the caption of Figure 3 it reads: “Majority of these star clusters are not expected to survive to the present, but will instead expand and tidally stripped apart to form the stellar disk and the halo. However, star clusters kicked out to large galactocentric radii may survive to become present day globular clusters.” So, what is the conclusion the reader should make? Can the authors kindly add references to the ejection mechanism mentioned? For example, such scenario is advocated in Forbes&Romanowsky 2023 and Adamo et al 2023 to explain the position of the GCs in the Sparkler. But huge literature is available on the topic. 1 to 2 references would suffice.

We thank the referee for pointing out that this figure was not explained in detail in the manuscript and that we have made a few assumptions which may not hold at high-z. Hence, we have removed this figure from the paper and replaced it with a new figure of surface density vs. size of star clusters (similar to Adamo et al. 2024). This figure compares the Firefly Sparkle star clusters to local GCs and to star clusters observed at high-z. We thank the referee for suggesting this improvement.

Reviewer Reports on the First Revision:

Referee #1 (Remarks to the Author):

I thank the authors for addressing my previous comments and for their revisions to the paper. I am generally satisfied with the changes they have made. I do have some remaining minor comments, which I list below, but otherwise I have no problem recommending this paper for publication.

Line 61: "Figure ??"

Lines 67-72: When talking about the distance between FF1- and other objects, specify that the physical distance is always given in the source plane, not just for FF-NBF.

Figure 2: The caption now seems to be saying that the top trace in the 2D spectrum is Slit 1, but this figure has been flipped from its original orientation, so it still is incorrect. Please check and double check the final version of this figure and clearly label Slit 1 and 2 in the 2D trace.

Method Section 2, 2nd paragraph (Line 261): This is more of a suggestion, but for the point-spread function, have the authors looked into using STARRED (Millon et al. 2024)? They have shown promising results for a test case similar to this one (on previous work by the author), so it may be worth at least commenting on.

Line 298: How is the position angle defined (I assume East of North)?

Method Section 2, 6th paragraph (Line 319): Why is the flux uncertainty determination procedure only done in the 10"x10" cutout and not the entire field? If the particular region of the cutout is somehow biased, this procedure wouldn't be able to account for that. 10"x10" is small enough that I don't think you are getting meaningful variance just by moving the model around within the cutout.

Figure 5: The caption says the PSF is shown as black circles in the lower right corner, but they are actually in the lower left corner.

Figure 7: The right panel is not sufficiently explained by the legend or the figure caption. Presumably the dashed lines and shaded regions are the measured line ratios. Where do the solid lines come from?

Figure 8: The contours in the corner plot (particularly the black contours) look like they need to be sampled more to get a smoother distribution, or the bins are too small.

Line 758: I do not think the "average magnification" of the FF-arc is a meaningful quantity because differential magnification can cause large gradients across the image. As I had previously suggested, why not give a "total" magnification of the FF-arc by taking the ratio of the flux of the reconstructed arc from the model (or the observed arc once the flux from the star cluster components are subtracted) to the de-lensed flux predicted by the same model?

Line 797: The intrinsic size of the clusters assumes they are spherical. This is probably a reasonable assumption, but for star clusters in the early stages of formation, is this consistent with theory/simulations?

Referee #2 (Remarks to the Author):

I have reviewed the updated version of the paper “Formation of a low-mass galaxy from star clusters in a 600 Myr old universe”, and I appreciate the work the authors have done to address my and the other referee’s comments. The paper reads well and clearly articulates the key results and their significance to the field. At this point, I believe this paper is largely ready for publication, however there are a couple minor points I would like to bring to the author’s attention.

- The fluxes listed in Table 4 do not appear to have been updated to reflect the updated photometry described in the author’s response. The fluxes appear to have not changed from the previous submission (then table 3). I’m assuming this is a version issue, since the photometry plotted in Figure 9 now appears in good agreement between the NIRCcam and NIRISS instruments.

- I suggest including citations to the JWST mission overview

(<https://ui.adsabs.harvard.edu/abs/2023PASP..135f8001G/abstract>), as well as the NIRISS, NIRCcam, and NIRSpec performance papers

(<https://ui.adsabs.harvard.edu/abs/2023PASP..135i8001D/abstract>;

<https://ui.adsabs.harvard.edu/abs/2023PASP..135b8001R/abstract>;

<https://ui.adsabs.harvard.edu/abs/2023PASP..135c8001B/abstract>; respectively) as a way of giving due credit to the many people who brought JWST to life.

Once the flux values in Table 4 are updated I will happily recommend this paper for publication in Nature.

Referee #3 (Remarks to the Author):

Dear Editor,

I thank the authors for their timely replies that have addressed the majority of my points. The revised manuscript is stronger and provide a coherent interesting analysis that clearly touches on the frontiers of research conducted with JWST.

I have only a few points here below for the authors to address. A list of minor typos follows.

i —> Reply to item 3.1 in the response to the referee report submitted by the authors. The point is about the use of SSPs vs. Non-parametric SFH for star cluster candidates. The authors motivate the use of non-parametric SFHs for star clusters because of 1. Possible residuals from the background, 2. referencing simulation of GMCs.

1. What is the fraction of light of the galaxy underlying the star cluster area? The authors discussed outshining. Since the diffuse light is subtracted, the residuals should be negligible. So this point is not truly strong.

2. all the references that the authors provide and in general simulations of isolated GMCs (e.g. <https://arxiv.org/pdf/2201.00882> and similars) or cluster formation in galaxies (e.g., see <https://ui.adsabs.harvard.edu/abs/2020ApJ...891....2L/abstract> and similars) find that massive star clusters form in several free fall times. This is not however a proof that star formation lasts over a long period in star clusters. When converting these free-fall times into absolute time scales it means <5 Myr in the region involving the star cluster (<5 pc). You can check the references I provide above, where the authors also quote ages and not free fall time units. Observationally, studies of color-magnitude diagrams of resolved stars in star clusters in MW, Magellanic Clouds, M31, M33 all show an extreme good agreement with isochrons of single ages. An important factor to notice is that all the studies that quote short burst of a few Myr are on a few parsec scales similar to the sizes observed in the manuscript. Only globular clusters with multiple Fe sequences, show discrete color sequences interpreted as age spread (Omega Cen for example). These age spread are observed in nuclear star clusters or remnant of dwarf nuclei, where the discrete ages can be explained either by episodic gas accretions toward the centre or merger of clusters.

I need to insist on the point that, in lack of evidence of long continuous SFH for star clusters (excluding nuclear star clusters which do not apply to the case currently studies), the authors should consider the SSP models as reference. I'm ok with including the non-parametric analysis and the fact the the authors point out that usters have a peak younger than 10 Myr, but I would appreciate that they also point out that the starting of the mass build up from 600 Myr comes from the initial condition that the authors assume and it is not physical for such small physical scales..

I would encourage the authors to rephrase:

+ Line 120-121 "SSP models are used for globular clusters, " —> "SSP models are typically used in studies of star clusters in the local universe, as both observational and numerical works finds that they can be approximated to single bursts,"

+ Sect 4 from lines 595 to 612. Those considerations apply for the FF galaxy but not to star clusters. The references there refer to galaxy studies not star clusters. I would encourage the authors to add the motivation of using SSPs as reference SFH for star clusters because of the lack of contrary evidence, star clusters appear to have remarkable short SFHs in both observations and simulations.

+Sect 4 from line 697 to 703. "By design, an SSP is biased towards this recent burst, whereas a non-parametric SFH can accommodate extended episodes of star formation. However, with our current data, we cannot distinguish between extended SFH in the star clusters and the contribution of light from the diffused arc." It is not by design, please reword. It is a simple light weighted effect, the same effect produces the pronounced recent increase in SFR for the non-parametric fits. The non-parametric SFHs accommodate extended star formation because as input it is set that star formation starts at the age of the universe, in this case 600 Myr. This assumption might be correct for galaxies but not for star clusters. The latter assumption also makes clusters significantly more massive, since stars have started forming by design 600 Myr ago. Finally, the data do not allow to differentiate between continuous and burst star formation in star clusters, but evidence from star and globular cluster observations in the local universe and simulations point toward the fact that at physical scales of a few parsec star clusters can be approximated by SSPs. Finally, when referring to the contamination from the galaxy diffuse light please add the term RESIDUAL diffuse light of the galaxy, since the diffuse light has mostly been subtracted.

ii —> Size measurement of FF-4. I got confused while reading the main text and the methods. Line 80 it says that FF-4 exhibit an elongated component, and an half light radius is reported. Then in lines 86 unresolved clusters have sizes upper limits below 0.02" and it concludes saying that 9 (why not 10?) clusters have size upper limits of 4-7 pc.. Should the reader deduce that FF-4 has a measured size, but it is not clear what. Then in Methods Section 2, it says that 9 clusters are PSF like, and FF-4 is unresolved. But doesn't being PSF-like implies being unresolved? Then in the caption of Figure 5 FF-4 is semi-resolved. I tried to understand reading the methods what exactly is going on. If I understand correctly all the cluster sizes, including FF-4, are assumed to be upper-limit and the HFWM of the PSF is used to derive the physical upper-limit. Is this correct? Or do the authors trust the PSF deconvolve size of 0.01" obtained for FF-4? If the former is correct, I would encourage the authors to say in the main text that all sizes are derived assuming the PSF radius as upper-limit. Then in the Methods the authors can specify that FF-4 appears visually resolved but the recovered size is smaller than the PSF and therefore not trustable. Or similar but please be consistent.

iii —> Line 177-184. I do not understand in what way the results of this manuscripts indicate that early galaxy formation exhibit pressure-regulated star formation. I am familiar with the works referenced. But in the current manuscripts there is no estimate of mid-plane pressure of the galaxy or SFR density or gas surface density. The authors do not even know what is the gas mass fraction of the system which is likely very high. The authors need to better motivate this sentence and link it to the results obtained in the study, otherwise this reads as a pure speculation disconnected from the rest of the manuscript.

iv —> Line 188-192. I am sorry but I miss the meaning of this sentence. What does it mean that early galaxy assembly occurs in dense star clusters rather than gas-rich disks?? Why the presence of dense star clusters in a low mass galaxy implies that galaxies do not assemble from gas-rich disks? If we understand galaxy assembly it is unavoidable that galaxies at these redshifts are gas rich (e.g. references provided by the authors). The latter component most likely provides the pressure in the disk to actually fragment and form such massive clusters. So I do not agree with the statement that the FF galaxy differs from conventional view. What I think is an important take away message from this study is that the physical conditions of the gas in early galaxies favour the formation of massive star clusters. Galaxies reported to have similar physical properties (extreme emission lines, high electron densities, inverse Balmer jump etc) most likely are powered by massive star clusters hosting a large number of (e.g. top-heavy IMF) very massive stars (e.g. Toppings et al 2024a, b, Cameron et al 2024, Katz et al 2024 recently appeared). So the FF galaxy does not differ from conventional views, it actually shows a snapshot of what is going on within early galaxies.

Other minor list of points:

1. Line 61, the link to Figure ref is broken
2. Line 131-133: when mentioning the stellar densities it would be good to refer to the plot in Figure 3.
3. Caption Figure 3. The references of the cluster samples are wrong and do not reflect the objects plotted. Please fix the right references to the high-redshift samples and add reference to the local sample. This figure is clearly adapted from Adamo et al 2024. It might be good practise to reference it in line 131-133.
4. The sentence line 137-140 is not clear. What do the authors are trying to say? Is the galaxy assembly history that will determine the future of these star clusters? Or the SFH assumptions? If the latter, why is that? I think a logical step is missing in the sentence, it could be good to revise adding in a few words what exactly is the link between SFH and survivability of the clusters. I guess here that the authors are talking about dissolution when mentioning that they might be globular cluster versus fragment of galaxies
5. Methods - Section 1. Please report what pixel scale the data have.
6. Methods - somewhere report the cosmology assumed
7. Line 510 - a 10^4 factor is missing in the reported temperature value.

Author Rebuttals to First Revision:

Referee #1 (Remarks to the Author):

I thank the authors for addressing my previous comments and for their revisions to the paper. I am generally satisfied with the changes they have made. I do have some remaining minor comments, which I list below, but otherwise I have no problem recommending this paper for publication.

1.1 Line 61: "Figure ??"

We thank the referee for spotting this typo which has now been fixed.

1.2 Lines 67-72: When talking about the distance between FF1- and other objects, specify that the physical distance is always given in the source plane, not just for FF-NBF.

We have clarified in that paragraph that "all distances quoted in the paper are projected distances in the source plane".

1.3 Figure 2: The caption now seems to be saying that the top trace in the 2D spectrum is Slit 1, but this figure has been flipped from its original orientation, so it still is incorrect. Please check and double check the final version of this figure and clearly label Slit 1 and 2 in the 2D trace.

We had forgotten to update the text in the caption and we thank the referee for spotting this mistake. We have fixed the caption labels and have also labeled the figure.

1.3 Method Section 2, 2nd paragraph (Line 261): This is more of a suggestion, but for the point-spread function, have the authors looked into using STARRED (Millon et al. 2024)? They have shown promising results for a test case similar to this one (on previous work by the author), so it may be worth at least commenting on.

We thank the referee for pointing out this interesting work. We will be re-analyzing this galaxy using the STARRED method in a future paper.

1.4 Line 298: How is the position angle defined (I assume East of North)?

The position angle is East of North and we have now mentioned it in Line 298 with the angle.

1.5 Method Section 2, 6th paragraph (Line 319): Why is the flux uncertainty determination procedure only done in the 10"x10" cutout and not the entire field? If the particular region of the cutout is somehow biased, this procedure wouldn't be able to account for that. 10"x10" is small enough that I don't think you are getting meaningful variance just by moving the model around within the cutout.

We repeated our process with 100 different 10" x 10" masked cutouts, sampling the entire field (excluding the edges), and injected the galaxy model into 10 random positions in each cutout. We

then calculated the uncertainty based on the 1000 refitted fluxes. While our uncertainty did not change significantly, this approach provides a more robust measurement of variance, and we appreciate the referee's suggestion. We have updated the Methods section to include this additional step.

1.6 Figure 5: The caption says the PSF is shown as black circles in the lower right corner, but they are actually in the lower left corner.

We thank the referee for spotting this oversight. We have fixed the caption.

1.7 Figure 7: The right panel is not sufficiently explained by the legend or the figure caption. Presumably the dashed lines and shaded regions are the measured line ratios. Where do the solid lines come from?

We have updated the caption with the following:

“The dashed lines with shaded regions are the measured line ratios. The solid lines denote predicted line ratios as a function of different electron temperatures from \texttt{PyNeb} \citep{Luridiana2015}.”

1.8 Figure 8: The contours in the corner plot (particularly the black contours) look like they need to be sampled more to get a smoother distribution, or the bins are too small.

We thank the referee for pointing this out. The contours are made using an atlas with 300,000 points, and we find that the features seem to be mostly a result of the extremely noisy likelihood surface from the spectrophotometric fits rather than the sampling, which doesn't result in an appreciably smoother surface even with a larger atlas.

1.9 Line 758: I do not think the "average magnification" of the FF-arc is a meaningful quantity because differential magnification can cause large gradients across the image. As I had previously suggested, why not give a "total" magnification of the FF-arc by taking the ratio of the flux of the reconstructed arc from the model (or the observed arc once the flux from the star cluster components are subtracted) to the de-lensed flux predicted by the same model?

We cannot calculate a "total" magnification as magnification is not additive. We have provided a flux-weighted average magnification of the object. Our method essentially does the same calculation that the referee is suggesting. We recomputed the magnification using the ratio of the flux of the model arc and the source plane model and arrived at the same magnification value.

The total fluxes have been computed from the FF-arc model on a similar pixel grid as the CANUCS images. The magnifications for all pixels with a flux higher than 1% of the maximum pixel flux were computed. The magnifications are combined to get the total magnification defined as the ratio between the area of the sum of all selected image pixels and the area of the sum magnification corrected pixels in the source plane.

1.10 Line 797: The intrinsic size of the clusters assumes they are spherical. This is probably a reasonable assumption, but for star clusters in the early stages of formation, is this consistent with theory/simulations?

We had experimented fitting with Gaussian and Sersic ellipses with the axis ratio as a free parameter for all the clusters and have gotten axis ratio = 1 (circular), albeit worse fits due to the high number of free parameters. Hence, we determined that except FF-4, all the clusters are unresolved point sources whose axis ratio we cannot determine with our data.

We have looked at several simulation papers which suggest that the shape of the star clusters during the early stages of formation depends on the conditions within the molecular clouds, such as density, temperature, and turbulence, and on the age of the clusters (Chen et al. 2004, Farias et al. 2019, Hui and Gnedin 2020). We will be exploring this avenue in a future paper and we thank the referee for this interesting question.

Referee #2 (Remarks to the Author):

I have reviewed the updated version of the paper “Formation of a low-mass galaxy from star clusters in a 600 Myr old universe”, and I appreciate the work the authors have done to address my and the other referee’s comments. The paper reads well and clearly articulates the key results and their significance to the field. At this point, I believe this paper is largely ready for publication, however there are a couple minor points I would like to bring to the author’s attention.

2.1 The fluxes listed in Table 4 do not appear to have been updated to reflect the updated photometry described in the author’s response. The fluxes appear to have not changed from the previous submission (then table 3). I’m assuming this is a version issue, since the photometry plotted in Figure 9 now appears in good agreement between the NIRCcam and NIRISS instruments.

We apologize for this mistake and thank the referee for spotting this oversight. We have updated the table with the new photometry. The figures were already updated.

2.2 I suggest including citations to the JWST mission overview (<https://ui.adsabs.harvard.edu/abs/2023PASP..135f8001G/abstract>), as well as the NIRISS, NIRCcam, and NIRSpec performance papers (<https://ui.adsabs.harvard.edu/abs/2023PASP..135i8001D/abstract> <https://ui.adsabs.harvard.edu/abs/2023PASP..135b8001R/abstract> <https://ui.adsabs.harvard.edu/abs/2023PASP..135c8001B/abstract> respectively) as a way of giving due credit to the many people who brought JWST to life.

Thank you making this suggestion and we should indeed do this in all JWST publications. We have added the references in our manuscript.

Once the flux values in Table 4 are updated I will happily recommend this paper for publication in Nature.

Referee #3 (Remarks to the Author):

Dear Editor,

I thank the authors for their timely replies that have addressed the majority of my points. The revised manuscript is stronger and provide a coherent interesting analysis that clearly touches on the frontiers of research conducted with JWST.

I have only a few points here below for the authors to address. A list of minor typos follows.

3.1 i —> Reply to item 3.1 in the response to the referee report submitted by the authors. The point is about the use of SSPs vs. Non-parametric SFH for star cluster candidates. The authors motivate the use of non-parametric SFHs for star clusters because of 1. Possible residuals from the background, 2. referencing simulation of GMCs.

3.1.1 What is the fraction of light of the galaxy underlying the star cluster area? The authors discussed outshining. Since the diffuse light is subtracted, the residuals should be negligible. So this point is not truly strong.

We agree with the referee that the residuals from the diffuse light should be minor in comparison with the flux in the individual clusters and/or the diffuse component, but it can vary from filter to filter and across the different clusters depending on surface brightness fluctuations since we have assumed a smooth profile for the diffuse arc. While the residual fluxes are generally in the 0.3-1.2% range in comparison with the modeled flux for any individual component, that can sometimes be enough to introduce systematics in estimating stellar populations (Wang et al. 2023, Narayanan, et al. 2023, Sun et al. 2024). Therefore, while not a strong point, we mention it for completeness.

3.1.2 all the references that the authors provide and in general simulations of isolated GMCs (e.g. <https://arxiv.org/pdf/2201.00882> and similars) or cluster formation in galaxies (e.g., see <https://ui.adsabs.harvard.edu/abs/2020ApJ...891....2L/abstract> and similars) find that massive star clusters form in several free fall times. This is not however a proof that star formation lasts over a long period in star clusters. When converting these free-fall times into absolute time scales it means <5 Myr in the region involving the star cluster (<5 pc). You can check the references I provide above, where the authors also quote ages and not free fall time units. Observationally, studies of color-magnitude diagrams of resolved stars in star clusters in MW, Magellanic Clouds, M31, M33 all show an extreme good agreement with isochrons of single ages. An important factor to notice is that all the studies that quote short burst of a few Myr are on a few parsec scales similar to the sizes observed in the manuscript. Only globular clusters with multiple Fe sequences, show discrete color sequences interpreted as age spread (Omega Cen for example). These age spread are observed in nuclear star clusters or remnant of dwarf nuclei, where the discrete ages can be explained either by episodic gas accretions toward the centre or merger of clusters.

I need to insist on the point that, in lack of evidence of long continuous SFH for star clusters (excluding nuclear star clusters which do not apply to the case currently studies), the authors should

consider the SSP models as reference. I'm ok with including the non-parametric analysis and the fact the the authors point out that users have a peak younger than 10 Myr, but I would appreciate that they also point out that the starting of the mass build up from 600 Myr comes from the initial condition that the authors assume and it is not physical for such small physical scales.

We thank the referee for their expertise on this topic and for providing the references above. We found the literature on this topic to be far from monolithic, but defer to the referee's expertise in star formation on small physical scales. For example,

Kim et al. (2018) find in FIRE that '*frequent mergers in high-redshift proto-galaxies could provide a fertile environment to produce long-lasting bound star clusters*', with a few clusters with $M^* \sim 10^5 M_{\text{sun}}$ surviving over 100 Myr.

Renzini (2023) state that '*This evidence argues for an extended stage of star formation within a forming globular cluster, during which stellar feedback was substantially ineffective and the nascent globular cluster was able to accrete processed gas from its surrounding, and efficiently convert it into successive stellar generations*'

Rusta et al. (2023) find that the individual star clusters can be self-consistently modeled as the remnants of galaxies that they call MW progenitors, with ages of ~150 - 200 Myr.

On the contrary, along with the references the referee provided, we also find,

Bastian et al. (2023) state '*Models of continuous star formation within clusters, lasting for hundreds of Myr, are ruled out at high significance (unless stellar initial mass function variations are invoked). Models for the (nearly instantaneous) formation of a secondary population within an existing first generation are not favoured, but are not formally discounted due to the finite sampling of age/mass-space.*'

Ma et al. (2023) find that '*The age spread of cluster stars is typically a few Myr and increases with cluster mass.*'

Given this, we have taken the referee's suggestion to adopt the SSP models as reference, and the **figures all show the SSP ages and masses when plotting the cluster properties**. We have updated the text to include the caveat of the modelling assumptions for the non-parametric SFHS (although dense basis does not require that SFR start at the big bang) being possibly unphysical on small spatial scales.

As the referee suggests, the crossing times we calculate for the clusters lie between 1-4 Myr depending on the cluster mass and size (see full list of values below), which coincide with the free-fall times in the Grudic et al. 2020 paper. .

'Observationally, studies of color-magnitude diagrams of resolved stars in star clusters in MW, Magellanic Clouds, M31, M33 all show an extreme good agreement with isochrons of single ages.' - while this is definitely true (and we have mentioned it in the revised text), **conditions at high**

redshift might be different and our current data can not statistically rule out a scenario where star formation takes place over a longer duration. Additionally, simulations have difficulty simulating star formation robustly at these epochs, and this is one of the first observations (the other being Cosmic Gems from Adamo et al. 2024, etc.) in this regime (low mass and high-z) to analyse resolved star clusters.

'Only globular clusters with multiple Fe sequences, show discrete color sequences interpreted as age spread (Omega Cen for example). These age spread are observed in nuclear star clusters or remnant of dwarf nuclei, where the discrete ages can be explained either by episodic gas accretions toward the centre or merger of clusters.' - the referee echoes our considerations on this. Since we have no concrete evidence that these are not nuclear star clusters or remnants of mergers from the recent past of the galaxy, we feel that it is important to model it with both methods until further follow-up observations can ascertain their true nature.

Individual cluster crossing times

SSP: ['3.83 Myr', '1.52 Myr', '2.54 Myr', '4.35 Myr', '2.90 Myr', '2.94 Myr', '3.29 Myr', '3.59 Myr', '4.20 Myr', '1.33 Myr']

DB: ['2.68 Myr', '3.21 Myr', '1.27 Myr', '1.08 Myr', '1.98 Myr', '1.27 Myr', '1.18 Myr', '1.45 Myr', '2.11 Myr', '2.01 Myr']

tage/tcross for the SSP model fits: [1.38, 5.55, 1.51, 0.97, 1.25, 0.99, 1.26, 1.13, 0.51, 2.68]

t50/tcross for the DB model fits: [36.17, 47.08, 71.32, 106.4, 42.71, 66.75, 81.92, 79.55, 40.25, 42.11]

I would encourage the authors to rephrase:

3.1.3 Line 120-121 "SSP models are used for globular clusters, " —> "SSP models are typically used in studies of star clusters in the local universe, as both observational and numerical works finds that they can be approximated to single bursts,"

We have made this change in the updated manuscript, and thank the referee for the suggestion.

3.1.4 Sect 4 from lines 595 to 612. Those considerations apply for the FF galaxy but not to star clusters. The references there refer to galaxy studies not star clusters. I would encourage the authors to add the motivation of using SSPs as reference SFH for star clusters because of the lack of contrary evidence, star clusters appear to have remarkable short SFHs in both observations and simulations.

We thank the referee for pointing this out, and have modified the sentence accordingly.

The new text now reads - *'for the possibility that the individual clusters are either nuclear star clusters or the remnants of dwarf galaxies that have recently merged with the system, the primary advantage of using Dense Basis is that non-parametric SFHs allow us to account for flexible stellar populations...'*,

and has been moved to after the text about star clusters fit with SSPs.

3.2 Sect 4 from line 697 to 703. “By design, an SSP is biased towards this recent burst, whereas a non-parametric SFH can accommodate extended episodes of star formation. However, with our current data, we cannot distinguish between extended SFH in the star clusters and the contribution of light from the diffused arc.” It is not by design, please reword. It is a simple light weighted effect, the same effect produces the pronounced recent increase in SFR for the non-parametric fits. The non-parametric SFHs accommodate extended star formation because as input it is set that star formation starts at the age of the universe, in this case 600 Myr. This assumption might be correct for galaxies but not for star clusters. The latter assumption also makes clusters significantly more massive, since stars have started forming by design 600 Myr ago. Finally, the data do not allow to differentiate between continuous and burst star formation in star clusters, but evidence from star and globular cluster observations in the local universe and simulations point toward the fact that at physical scales of a few parsec star clusters can be approximated by SSPs. Finally, when referring to the contamination from the galaxy diffuse light please add the term RESIDUAL diffuse light of the galaxy, since the diffuse light has mostly been subtracted.

We acknowledge the referee’s point. We have removed the relevant sentence and replaced it with the following text:

‘It is important to note that there are two scenarios for the clusters seen in the Firefly Sparkle, for which the two different SED modeling assumptions can introduce biases. For the scenario where they are star clusters, the $\backslash db \}$ fits tend to produce extended episodes of star formation which are not physical at small physical scales as evidenced by studies of star and globular cluster observations in the local universe and simulations. On the other hand, if the clusters are nuclear star clusters or remnants of dwarf galaxies that merged with the system in the past, the SSP fits will pick up only the most recent burst of star formation, providing the light-weighted age, which tends to often be younger than the mass weighted age.’

3.3 —> Size measurement of FF-4. I got confused while reading the main text and the methods. Line 80 it says that FF-4 exhibit an elongated component, and an half light radius is reported. Then in lines 86 unresolved clusters have sizes upper limits below 0.02” and it concludes saying that 9 (why not 10?) clusters have size upper limits of 4-7 pc.. Should the reader deduce that FF-4 has a measured size, but it is not clear what. Then in Methods Section 2, it says that 9 clusters are PSF like, and FF-4 is unresolved. But doesn’t being PSF-like implies being unresolved? Then in the caption of Figure 5 FF-4 is semi-resolved. I tried to understand reading the methods what exactly is going on. If I understand correctly all the cluster sizes, including FF-4, are assumed to be upper-limit and the HFWM of the PSF is used to derive the physical upper-limit. Is this correct? Or do the authors trust the PSF deconvolve size of 0.01” obtained for FF-4? If the former is correct, I would encourage the authors to say in the main text that all sizes are derived assuming the PSF radius as upper-limit. Then in the Methods the authors can specify that FF-4 appears visually resolved but the recovered size is smaller than the PSF and therefore not trustable. Or similar but please be consistent.

We apologize for the confusion and have updated the text according to suggestion. In the manuscript it now states:

“We derived an upper limit on the half-light radii ($R_{\text{eff}} < 0.02$), which is 0.5 times the FWHM of the PSF in the F115W image, for all ten clusters, including FF-4, whose deconvolved size is smaller than the PSF of the image.”

3.4 —> Line 177-184. I do not understand in what way the results of this manuscripts indicate that early galaxy formation exhibit pressure-regulated star formation. I am familiar with the works referenced. But in the current manuscripts there is no estimate of mid-plane pressure of the galaxy or SFR density or gas surface density. The authors do not even know what is the gas mass fraction of the system which is likely very high. The authors need to better motivate this sentence and link it to the results obtained in the study, otherwise this reads as a pure speculation disconnected from the rest of the manuscript.

We agree with the referee and have reworded the sentence to read as follows:

‘With massive star clusters exhibiting high surface density, low metallicity, high electron temperature, and hints of a top-heavy IMF, the Firefly Sparkle exhibits the hallmarks of star formation in extreme environments, consistent with scenarios like pressure-regulated, feedback dominated star formation (Ostriker2020, Ellison2024, Kim2024), though further observations of the gas mass are needed to ascertain this.’

3.5 —> Line 188-192. I am sorry but I miss the meaning of this sentence. What does it mean that early galaxy assembly occurs in dense star clusters rather than gas-rich disks?? Why the presence of dense star clusters in a low mass galaxy implies that galaxies do not assembly from gas-rich disks? If we understand galaxy assembly it is unavoidable that galaxies at these redshifts are gas rich (e.g. references provided by the authors). The latter component most likely provides the pressure in the disk to actually fragment and form such massive clusters. So I do not agree with the statement that the FF galaxy differs from conventional view. What I think it is an important take away message from this study is that the physical conditions of the gas in early galaxies favour the formation of massive star clusters. Galaxies reported to have similar physical properties (extreme emission lines, high electron densities, inverse Balmer jump etc) most likely are powered by massive star clusters hosting a large number of (e.g. top-heavy IMF) very massive stars (e.g. Toppings et al 2024a, b, Cameron et al 2024, Katz et al 2024 recently appeared). So the FF galaxy does not differ from conventional views, it actually show a snapshot of what is going on within early galaxies.

We agree with the referee’s point and have reworded the sentence to denote early disks rather than gas-rich disks. As the referee suggests, the system is likely very gas rich, and the point we wanted to make was that the majority of star formation in the galaxy tends to be occurring in the dense star clusters rather than in a rotation-supported disk like a lot of studies suggest. This is not differing from conventional views, in that many studies including the ones cited in the paper show this to happen in simulations, but differs from the body of literature that claim that disks form at extremely high redshifts and form the basis for early galaxy formation. The revised sentence reads:

“The Firefly Sparkle suggests that early galaxy assembly can occur via dense star clusters as well (Iza et al, 2023; Feng et al, 2015; Nepal et al, 2024).”

Other minor list of points:

3.6 Line 61, the link to Figure ref is broken

We thank the referee for spotting this typo which has now been fixed.

3.7 Line 131-133: when mentioning the stellar densities it would be good to refer to the plot in Figure 3.

We have added the reference.

3.8 Caption Figure 3. The references of the cluster samples are wrong and do not reflect the objects plotted. Please fix the right references to the high-redshift samples and add reference to the local sample. This figure is clearly adapted from Adamo et al 2024. It might be good practise to reference it in line 131-133.

We thank the referee for spotting this mistake. We have updated the references. We have also added the reference to Adamo et al 2024.

3.9 The sentence line 137-140 is not clear. What do the authors are trying to say? Is the galaxy assembly history that will determine the future of these star clusters? Or the SFH assumptions? If the latter, why is that? I think a logical step is missing in the sentence, it could be good to revise adding in a few words what exactly is the link between SFH and survivability of the clusters. I guess here that the authors are talking about dissolution when mentioning that they might be globular clusters versus fragments of galaxies.

We have modified the sentence in line with the other changes suggested by the referee to now read as follows:

'The precise nature of the clusters depends on the interpretation of their star formation histories. If seen as star clusters, their masses lead to crossing times of 1-4 Myr. Combined with their age estimates, this puts them at $t_{\text{age}}/t_{\text{cross}} \sim 1-2$, which indicates that they are marginally bound. On the other hand, if they are nuclear star clusters or the remnants of dwarf galaxies that have previously merged with the system, their ages are consistent with having survived several crossing times and are likely to remain bound until ejected from the system or integrated into the nucleus.'

3.10 Methods - Section 1. Please report what pixel scale the data have.

We have added these sentences in Method Section 1.

"The images have a pixel scale of 0."04 /pixel."

3.11 Methods - somewhere report the cosmology assumed.

We have added these sentences in Method Section 2.

“We assume a cosmological model based on the Λ CDM framework, with the Hubble constant $H_0 = 70$ km/s/Mpc. The matter density parameter $\Omega_m = 0.3$, and the dark energy density parameter $\Omega_\Lambda = 0.7$. All magnitudes are expressed using the AB magnitude system.”

3.12 Line 510 - a 10^4 factor is missing in the reported temperature value.

We thank the referee for spotting this typo which has now been fixed.

Reviewer Reports on the Second Revision:

Referee #3 (Remarks to the Author):

I thank you the authors for their answers and congratulate them on the publication of this interesting work.